# Uses of Scanning Electrochemical Microscopy (SECM) for the Characterization with Spatial and Chemical Resolution of Thin Surface Layers and Coating Systems Applied on Metals: A Review

**Juan J. Santana [1,\*], Javier Izquierdo [2,3] and Ricardo M. Souto [2,3,\*]**

[1] Department of Process Engineering, University of Las Palmas de Gran Canaria, 35017 Las Palmas de Gran Canaria, Spain

[2] Department of Chemistry, Universidad de La Laguna, 38200 La Laguna, Tenerife, Spain; jizquier@ull.edu.es

[3] Institute of Material Science and Nanotechnology, Universidad de La Laguna, 38200 La Laguna, Tenerife, Spain

[\*] Correspondence: juan.santana@ulpgc.es (J.J.S.); rsouto@ull.es (R.M.S.)

**Abstract:** Scanning Electrochemical Microscopy (SECM) is increasingly used in the study and characterization of thin surface films as well as organic and inorganic coatings applied on metals for the collection of spatially- and chemically-resolved information on the localized reactions related to material degradation processes. The movement of a microelectrode (ME) in close proximity to the interface under study allows the application of various experimental procedures that can be classified into amperometric and potentiometric operations depending on either sensing faradaic currents or concentration distributions resulting from the corrosion process. Quantitative analysis can be performed using the ME signal, thus revealing different sample properties and/or the influence of the environment and experimental variables that can be observed on different length scales. In this way, identification of the earlier stages for localized corrosion initiation, the adsorption and formation of inhibitor layers, monitoring of water and specific ions uptake by intact polymeric coatings applied on metals for corrosion protection as well as lixiviation, and detection of coating swelling—which constitutes the earlier stages of blistering—have been successfully achieved. Unfortunately, despite these successful applications of SECM for the characterization of surface layers and coating systems applied on metallic materials, we often find in the scientific literature insufficient or even inadequate description of experimental conditions related to the reliability and reproducibility of SECM data for validation. This review focuses specifically on these features as a continuation of a previous review describing the applications of SECM in this field.

**Keywords:** scanning electrochemical microscopy; corrosion protection; coating degradation; corrosion inhibitor films; electrochemical activity; microelectrode

## 1. Introduction

Corrosion involves the destructive oxidation of metals and non-metallic materials, which causes degradation of their function as a result of exposure of the materials to environments that are aggressive to them. Most of the materials used in our society require a contribution of energy for their extraction and industrial production in the desired chemical state and form. Therefore, there is a great thermodynamic tendency for these materials to return to their original, more stable state—hence corrosion is ultimately an unavoidable process. However, the use of proper corrosion control methods significantly slows down the rate at which the corrosive phenomenon occurs, making it possible to increase the useful life of the material, thereby reducing the impact on the environment

by a lower consumption of materials, namely by reducing the rate of replacement, as well as by the lower contamination due to degradation products.

The degradation reactions that occur in metallic materials exposed to the atmosphere or to an aqueous medium have their electrochemical origins in common. That is to say, they are produced by the development of electrochemical microcells on the surface of the corroding material, with dimensions typically within the micrometric and even submicrometric range in their beginnings. Thus, the anodic half-reaction (namely the oxidation of the metal) and the cathodic half-reaction (i.e., the oxygen reduction reaction in most aqueous environments) occur at different locations on the surface, forming highly localized microcells. As a result, several surface phenomena emerge, such as the formation of passive layers, local electric fields are generated, changes in the surface conductivity, or variations in the rate of electron transfer reactions, which, along with the possible coupling of homogeneous-phase reactions in solution, generation of local electric fields, changes in pH, etc., ultimately account for the degradation pathways, as shown in Figure 1. As most of the knowledge of corrosion mechanisms has been gathered using conventional electrochemical techniques that are surface averaging methods, little or no information is currently available on reactivity at sites of corrosion initiation or at small defects in surface layers and films. This situation is a major drawback for the development of effective corrosion protection technologies, which often can only attempt to minimise the extent of corrosion once it has started, as the actual mechanisms related to its initiation remain mostly unknown.

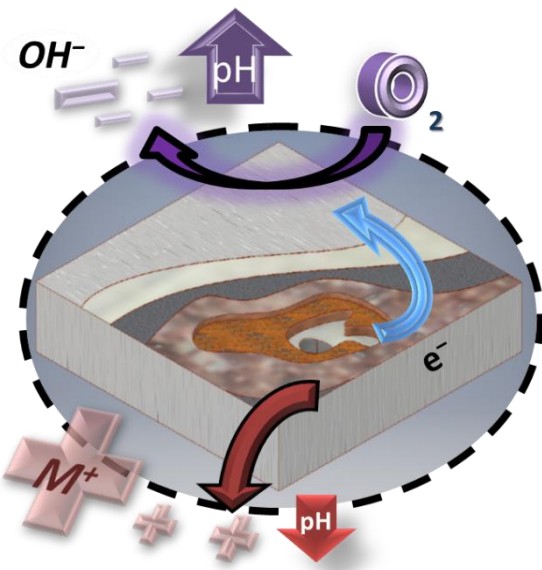

**Figure 1.** Sketch depicting the formation of localized microcells, the subsequent flow of ionic species in the electrolyte, and the pH changes associated with the onset of corrosion of a metal immersed in an alkaline or neutral aqueous medium.

The most effective and widely employed anticorrosion protection method consists of the application of organic coatings on metals. Organic coatings mainly provide a physical barrier against the access of water and ions to the metal surface, preventing the onset of electrochemical reactions there. However, there is no completely impermeable coating to these chemical species and therefore a certain electrochemical activity will find its origin in small defects, which are invariably present, but more notably in larger defects which are produced by the action of the environment (e.g., scratches, particle impacts, etc.) or on the cut edges of coated metals [1]. The addition of corrosion inhibitors could prolong the life of the coated material by inhibiting corrosive processes at these defects and cut edges, provided a steady supply of them is maintained when added to the coating as pigments.

This supply is not sustained in a controlled manner in typical anticorrosion coating formulation, so they are released by the coating continuously, eventually ceasing to be available for the protection of the material, and polluting the environment. A better concept is that the coatings could be functionalized for the release of the inhibitor only when corrosion has started, in order to stop the degradation processes and to heal the defect formed [2–4]. Such "smart" or self-repairing coatings, modified with additives and pigments specially designed for the mitigation of corrosive activity, would not release large amounts of chemicals into the environment, while providing more effective corrosion protection.

Although electrochemical techniques provide powerful tools to study interfacial reactions, especially corrosion processes, the conventional methods lack spatial resolution and provide limited information about electrochemical behaviour at sites of corrosion initiation or defects and cut edges. In fact, processes occurring at these sites are highly localized and heterogeneously distributed throughout the system (cf. Figure 1), and methods with spatial resolution that can acquire real-time data are needed to obtain relevant information about the underlying reaction mechanisms. In the last three decades, significant progress has been made in the knowledge and analytical monitoring of the origins of corrosion by means of electrochemical measurements at micrometric and sub-micrometric scale. Some methods have been devised to considerably reduce the number of degradation events that can occur simultaneously on a given substrate, even resulting in the electrochemical signal measured in the system coming from distinct events. This is achieved by miniaturizing the measurement cell [5] or the sample under study using individual microelectrodes [6] and microelectrode arrays [7]. The measured signal always represents an average of all events occurring on the exposed substrate, but transients related to individual breakdown events or localized sites can be distinguished above the background signal. In this way, it is possible to resolve the different steps of the breakdown process [5–8], although the use of ex situ optical, electron, or atomic force microscopies is necessary to correlate the electrochemical findings with specific features of the object studied [9–11].

Scanning microelectrochemical techniques provide an alternative for the study of protection methods and degradation reactions with spatial selectivity, especially with regards to the initiation and subsequent propagation of the latter [12–15]. Conceptually, these methods can be considered scanning probe microscopies (SPMs) in which microelectrodes are used as sensing probes in near-field scanning configurations, leading to the monitoring of electrochemical systems that are both chemically and spatially resolved for the surfaces under investigation. In fact, SPM is a branch of microscopy where the pixels of the eventual image are obtained by sequentially implemented local measurements. The main three groups of scanning microelectrochemical techniques are based on the measurement of local potential fields (namely, the scanning reference electrode technique, SRET, and the scanning vibrating electrode technique, SVET) [12], the measurement of local electrochemical impedances (i.e., the localized electrochemical impedance spectroscopy, LEIS [15]), and the electrochemical operation of microelectrodes in the scanning electrochemical microscope (SECM) [14]. Among them, the only technique capable of obtaining chemical resolution and specificity is SECM, since the measuring probe can be configured to specifically monitor a certain chemical species.

Since its introduction in 1989 by Engstrom's [16] and Bard's [17] research groups, SECM has found increasing application in various scientific fields, namely Chemistry, Materials Science, Chemical Engineering, and Corrosion Engineering, among others [18]. As result, the scientific production using SECM has been steadily growing over the years and reporting applications in different research areas. In particular, SECM has also been used for the characterization of organic and inorganic coatings applied on metals, where this technique has found application for microscopic chemical imaging, the measurement of physicochemical constants and coefficients, and as a micromachining tool [19]. A relevant review addressing strategies and operation modes in SECM for the investigation of corrosion processes in metals and their alloys was published by Payne et al. in 2017 [14].

The latter work contained a brief section about the uses of SECM in the investigation of corrosion reactions in coated metals including a summary of different corrosion protection schemes classified by kind of protection and metal.

In 2016, a comprehensive review of experimental parameters in SECM was published by Polcari et al., covering applications in different fields [18], whereas Zoski focused on the use of SECM for surface reactivity characterization, emphasizing novel operation modes, as well as the detection and quantification of metal oxides of the materials used for tip fabrication [20]. Unfortunately, no similar effort has been made in previous reviews on the application of SECM to the study of corrosion processes for covering key experimental parameters of the measurements for data such as the potential applied to the tip, the composition of the measuring solution, tip stability and dimensions, or the eventual effect of redox mediator conversion at the tip on the actual corrosion process under investigation.

The present review addresses the different modes of operation in SECM and their application to the study of coatings and metal-coating systems, with a detailed breakdown of experimental aspects. This work effectively updates and extends our previous report where the operation modes available in SECM for the study of degradation processes in coated metals were described and illustrated [1], also including a review of the experimental parameters involved in the measurements that constitute analytical figures of merit for the comparison and quantitative evaluation of SECM data [21].

## 2. Experimental Design for SECM Operation

There are several important factors to be considered when designing a SECM experiment. The most important ones are the nature and geometry of the tip that will determine the spatial and chemical resolutions of the measurement, as well as the type of substrate and mediator, and the solvent to be employed. A review of the main experimental design factors related to the application of SECM to the analysis of coating systems is summarized in the next sections. All the information is organized in tables in order to simplify the presentation of the available resources and their easier comparison. We also provide a brief description of the main aspects related to the features described in the next subsections. It must be noted that method validation will not be discussed here as it is beyond the reach of this work, but a relevant review on the topic has been recently presented by Izquierdo et al. [21].

### 2.1. SECM Instrumentation

SECM is a scanning probe microscope (SPM) technique based on electrochemical principles. The movement of a microelectrode (ME) in close proximity to the interface under study allows the application of various experimental procedures that can be classified into amperometric and potentiometric operations depending on either sensing faradaic currents or probe potential values due to concentration distributions resulting from the corrosion process, as sketched in Figure 2. In addition, alternating current signals can be applied to the ME, leading to AC-operation modes.

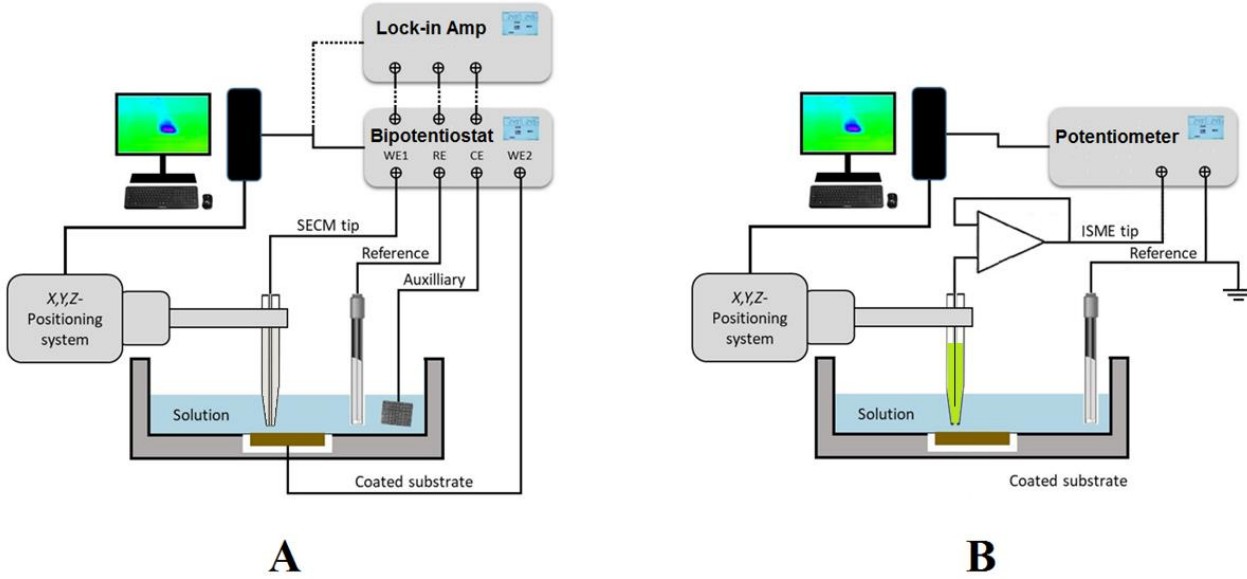

**Figure 2.** Sketches depicting the SECM set up and electrode connections for (**A**) amperometric and AC operations; and (**B**) potentiometric operation.

As sketched in Figure 2A, amperometric SECM operation is performed in a small electrochemical cell constituted by the tip, the counter electrode, the reference electrode, the substrate, and the solution. A bipotentiostat completes the SECM electrochemical setup together with the electrochemical cell, and it can be used to independently control the potential (bias) of the tip and the substrate, although the latter can also be left unbiased at its spontaneous corrosion potential in the environment. Next, a micropositioner is driven by stepper motors or piezoelectric elements to achieve the movement of the tip in the *X*, *Y*, and *Z* directions for exploring the substrate with submicrometric resolution. The experimental setup is completed with the interface, display system, and computer that records the current at the tip (and eventually the substrate when polarized by the bipotentiostat) as a function of tip position or the potential of the corresponding working electrode.

AC modes are available by attaching a lock-in amplifier or a frequency response analyser (FRA) to the bipotentiostat, as shown in Figure 2A. 4D AC-SECM mode involves the electrochemical imaging of the AC components of the current signal flowing at the tip (i.e., admittance and phase angle) [22], whereas impedance spectra can be generated at the scanning electrochemical impedance microscope (SEIM) by combining the local current and potential signals [23]. Research applications of these techniques to characterize thin surface layers and coatings on metals are summarized in Table 1.

**Table 1.** Selected summary of successful applications of AC modes in SECM for the investigation of thin surface layers and coatings on metals.

| Technique | Application | Reference |
|---|---|---|
| AC-SECM | Visualisation of pin holes on lacquered tinplate | [24,25] |
| AC-SECM | Imaging of a scratch in polymer-coated galvanized steel | [26] |
| AC-SECM | Visualization of the adsorption of corrosion inhibitor layers on copper | [27,28] |
| AC-SECM | Definition of a characteristic threshold frequency during adsorption of corrosion inhibitor layers on copper | [29] |
| AC-SECM | Water uptake and early coating swelling in coil coated steel | [30] |
| AC-SECM | Holiday produced in a thin epoxyphenolic varnish applied on tinplate | [31,32] |
| SEIM | Visualization of the adsorption of corrosion inhibitor layers on copper | [23,33] |

| SEIM | Alumina layers on aluminium | [34] |
| AC-SECM | Self-healing performance of smart coatings loaded with corrosion inhibitors | [35,36] |

Alternately, potentiometric operation can be performed by measuring local potential signals in a two-electrode cell configuration, as sketched in Figure 2B. In this case, a high input impedance operational amplifier must be introduced between the electrode connections in the electrochemical cell before they are driven to the bipotentiostat or the potentiometer unit employed as electrochemical interface [37]. Ion-selective microelectrodes (ISMEs) are used as SECM probes instead of the active tip surfaces employed in amperometric and AC modes. Although potentiometric SECM, also known as the scanning ion selective electrode technique (SIET) in some publications, has usually been performed using two separate electrodes in the electrochemical cell, as sketched in Figure 2B, it was demonstrated that such an arrangement contributes to big uncertainties in the measurement of local potential values in systems undergoing corrosion reaction, due to the high electrical fields developed in the electrolyte by galvanic pair systems [38]. The use of internal reference electrodes built inside the ion-selective electrode tip should be mandatory in order to overcome this reported limitation [39,40]. Another limitation arises from the rather slow equilibration time required to establish a stationary Donnan potential in the ion-selective membrane of the ISME, effectively limiting the resolution of the chemical images that can be recorded using this operation mode, which is often reduced to a few 2D line scans [41], although new imaging procedures involving the construction of 3D pseudo-maps have recently become available [42].

A summary of the SECM instruments and the analytical figures of merit necessary for adequate description of the experiments and reproducibility are reviewed elsewhere [21].

### 2.2. Tips Used for Amperometric Operation

Tips employed in amperometric SECM are active microelectrodes (MEs) of a critical dimension below 25 μm for conditioning the mass transport of a redox species from the solution toward the electrode, which are used to characterize coatings and/or thin surface layers applied on metals without requiring any additional modification [43]. In SECM, the most employed tip is built using a platinum wire of different diameters, followed by gold and carbon microwires or fibres, although antimony- and iridium-based tips have been occasionally employed due to the dual amperometric/potentiometric potential of their oxides [44,45].

For the tip fabrication, usually a metal microdisk (e.g., Pt) is sealed into a glass capillary and tapered to a conical shape. Then, it is polished with graded alumina powder (or similar) of different sizes in order to expose a disk-shaped electrode with an active surface for the redox reaction over its surface and is fully characterized by recording a cyclic voltammogram of a known electroactive species (e.g., redox mediator) added to the test solution, as it is exemplified in Figure 3 for the case of the oxidation of ferrocene-methanol on a Pt ME.

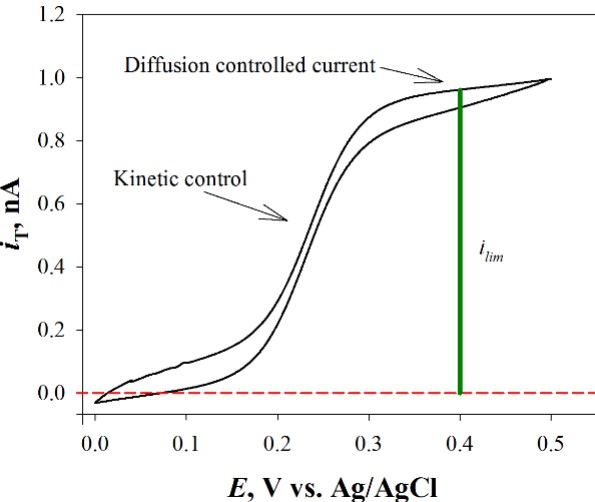

**Figure 3.** Cyclic voltammetry curve for ferrocene-methanol (i.e., the most employed redox mediator in the literature) on a Pt ME.

The variation of the measured current ($i_T$) at the surface of the ME with potential makes it possible to obtain relevant data on the active surface and the tip geometry, as well as the value of the stationary diffusion current, $i_{lim} = i_{T,\infty}$, and the operative potential range that is available at the tip. This stationary current is controlled by the electrochemical characteristics of the redox species (namely its diffusion coefficient and concentration) together with the geometrical factors of the tip [46]. The latter are accounted for by the radius of the electroactive surface of the electrode (*a*, in cm), and a geometric coefficient, *β*, that depends on the ratio between the diameters of the active disk electrode (2*a*) and of the insulating shaft built around it (*S*) that is named the $R_g$ value [46]. This stationary current is measured for the tip placed in the bulk of the solution, and it is effectively observed when the probe is positioned at a distance from the substrate 10 times greater than *a*.

To compare information from different measurements, SECM data are usually plotted using normalized quantities for the current, *I*, and for the tip-substrate distance, *L*. These parameters are obtained as the ratios of the current measured at the tip at some distance from the substrate, *d*, to the stationary current ($I = i_T/i_{T,\infty}$), and to the tip radius ($L = d/a$), respectively.

Although glass-embedded metallic Pt tips, with a clear predominance of the tips of 10 to 25 μm in diameter, are the most used in this field, in some cases Pt coated with parylene C [47] and a Pt/IrO$_x$ tip [48], both of 25 μm diameter, have been employed. Although tip diameters up to 100 μm have been reported for SECM application [49–51], it must be taken in account that they are too big for applying the analysis tools developed for microelectrode configurations, which are described in Section 3. Other materials used have been Au disks (5 and 25 μm of diameter) [52], and boron-doped diamond (BDD) films on tungsten wires [53].

### 2.3. Redox Mediators

A redox mediator is an electroactive molecule, atom, or ion that can be reduced or oxidized. Mediators are classified as direct and indirect redox mediators depending on whether the species is already present in the solution (e.g., O$_2$) or has to be added to the solution to perform the experiment (e.g., ferrocene-methanol, FcMeOH). The latter is needed to monitor either insulating or poorly-conductive substrates, thus reflecting only topographical and/or chemical reactivity changes along a surface, respectively, as sketched in Figure 4. Redox mediators may also be added for precise tip positioning of the tip relative to the substrate by recording the changes in the tip current while

approaching the substrate (see Section 3.1), as well as to locate sites of different reactivity on the substrate (cf. Figure 4). Normally, when a redox mediator species is added to the solution, a low concentration (approximately in mM values) is used. However, more strictly, the concentration of the mediator must be chosen by taking into account its reaction rate at the substrate. Besides, it may be necessary to adjust the pH.

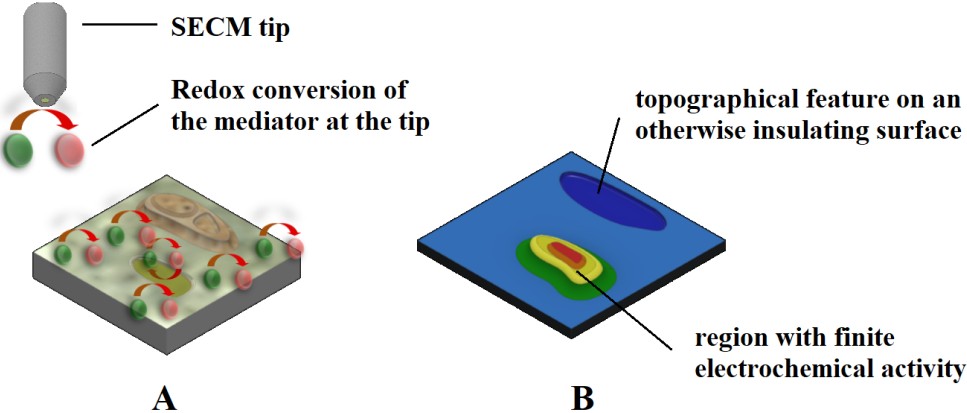

**Figure 4.** Sketches depicting: (**A**) the redox conversion of an electrochemical mediator at the amperometric tip of SECM over a mostly insulating surface, whereas the redox mediator regeneration solely occurs in a region exhibiting electrochemical reactivity; and (**B**) the current measured at the tip, which contains information on the topography of the sample (i.e., blue color palette) eventually coupled with a region showing a heterogeneous chemical reactivity distribution (red and yellow color palette).

The selection of a certain redox mediator for a given experiment is a critical issue for a successful experiment and depends on several factors, such as:

- The nature of the sample studied;
- The nature of the mediator (chemical stability, redox potential, photostability, toxicity, thermal stability, and solubility in the solution to be tested); and
- The mode of operation to use in SECM.

A broad classification of redox mediator systems for the investigation of electrochemical corrosion processes can be made by considering whether the species is added to the test environment for imaging (ideal redox mediator system) or whether it is a certain chemical species that participates in the corrosion mechanism, although the latter frequently exhibits poorly reversible or even irreversible electron transfer reactions (for instance, the electroreduction of molecular oxygen) and highly variable concentration ranges. Since corrosion reactions on coated metals often expose insulating or poorly conductive layers to the electrolytic phase in which the SECM tip is moved, the use of corrosion-related mediators is limited to systems presenting either defects or cut edges, or low efficiencies of inhibition. In contrast, the addition of redox mediators that exhibit fast, simple, and highly reversible electron transfer reactions at the tip is preferred in the case of non-defective and barrier-type layers and coatings, where the initiation of the corrosion reactions occurs in the buried interface formed by the metal and the surface layer. In this case, the information collected on the degradation process is obtained by observing morphological and topographical changes on the outer surface of the coating or film. An intermediate case occurs during the formation of inhibiting layers on metals by adsorption, because there is a gradual transition from an electrochemically active surface to a (quasi) insulating surface, making it possible to follow the time course of the charge transfer reaction at the surface [54]. In this case, the main care must relate to the selection of a mediator having a redox potential close to the corrosion potential of the substrate, in order to minimize the effects of polarization of the substrate by the redox couple.

Although at least 133 molecules have been used or investigated for amperometric operation in SECM since its inception [18], Table 2 shows that only a few of them have been used for SECM characterization of thin surface layers and coatings on metals.

**Table 2.** Redox mediators used in the amperometric SECM characterization of thin surface layers and coatings on metals.

| Mediator | Abbreviation | Redox Reaction | Redox Potential (V vs. NHE) | Reference |
|---|---|---|---|---|
| Azobenzene | AB | $AB + e^- \rightarrow AB^{\bullet-}$ | +1.378 | [18] |
| Benzoquinone/hydroquinone | BQ/HQ | $BQ + 2H^+ + 2e^- \rightarrow HQ$ | −0.278 | [55] |
| Decamethylferrocene | DcMeFc | $[DcMeFc]^+ + e^- \rightarrow DcMeFc$ | +0.261 | [56] |
| Dimethylamino-methylferrocene | DMAMFc | $[DMAMFc]^+ + e^- \rightarrow DMAMFc$ | +0.551 | [57] |
| Ferrocene | Fc | $Fc^+ + e^- \rightarrow Fc$ | +0.665 | [58] |
| Ferrocenemethanol | FcMeOH | $[FcMeOH]^+ + e^- \rightarrow FcMeOH$ | +0.500 | [59] |
| Hexaammineruthenium (III) | $[Ru(NH_3)_6]^{3+}$ | $[Ru(NH_3)_6]^{3+} + e^- \rightarrow [Ru(NH_3)_6]^{2+}$ | −0.059 | [60] |
| Hexacyanoferrate (III) | $[Fe(CN)_6]^{3-}$ | $[Fe(CN)_6]^{3-} + e^- \rightarrow [Fe(CN)_6]^{4-}$ | +0.491 | [61] |
| Hydrogen | $H_2$ | $2H^+ + 2e^- \rightarrow H_2$ | 0.000 | [18] |
| Hydrogen peroxide | $H_2O_2$ | $O_2 + 2H^+ + 2e^- \rightarrow H_2O_2$ | +0.670 | [62] |
| Iodide | $I^-$ | $I_3^- + 2e^- \rightarrow 3I^-$ | +0.963 | [63] |
| Iodine | $I_2$ | $I_2 + 2e^- \rightarrow 2I^-$ | +0.532 | [64] |
| Iridium chloride | $IrCl_6$ | $[IrCl_6]^{2-} + e^- \rightarrow [IrCl_6]^{3-}$ | +0.870 | [65] |
| Iron | Fe | $Fe^{3+} + e^- \rightarrow Fe^{2+}$ | +0.772 | [66] |
| Methylviologen | MV | $MV^{2+} + e^- \rightarrow MV^+$ | −0.446 | [67] |
| 4-nitrobenzonitrile | 4NB | $4NB + e^- \rightarrow 4NB^{\bullet-}$ | +0.659 | [68] |
| Oxygen | $O_2$ | $O_2 + 2H_2O + 4e^- \rightarrow 4OH^-$ | +0.401 | [62] |
| Oxygen | $O_2$ | $O_2 + e^- \rightarrow O_2^-$ | −0.498 | [69] |
| Tetramethyl-p-phenylenediamine | TMPD | $TMPD^+ + e^- \rightarrow TMPD$ | +0.258 | [70] |
| Tetracyanoquinodimethane | TCNQ | $TCNQ + e^- \rightarrow TCNQ^-$ | +0.322 | [18] |
| Tetrathiafulvalene | TTF | $TTF^{2+} + e^- \rightarrow TTF^+$<br>$TTF^+ + e^- \rightarrow TTF$ | +0.593<br>+0.943 | [64] |

### 2.4. Tips Used for Potentiometric Operation

Passive tips (ISME) are employed in potentiometric SECM. Since ion activities are detected using ISME tips, without being consumed during the measurement, no interaction must occur with the sample surface. Although chemical selectivity is thus envisaged compared to amperometric operation, the selectivity of the probe is not always sufficiently high and it is necessary to be investigated in regards to other ions present in the system [71]. Additionally, longer acquisition times are required for SECM measurement due to longer response times of the probes. When scanning rates similar to those typical for amperometric SECM operation are employed, the recorded images may exhibit significant aberration effects. Nevertheless, significant improvement has been achieved by combining dedicated scanning routines with mathematical deconvolution procedures [72,73].

Potentiometric probes used in SECM can be classified into two broad categories, namely metal-based microsensors [74] and reference microelectrodes [75]. The first class of potentiometric microsensors take advantage of the passive properties of certain metal oxides that are primarily sensitive mainly to pH changes in the environment, such as antimony and iridium. Although the narrow potential range of stability of its metallic state in electrolyte solutions precludes its use as an electrode material with voltammetric techniques, it encompasses the potential range of $O_2$ electroreduction and therefore can be

used in SECM for precise probe positioning [76,77], which is a typical limitation of most passive potentiometric probes. In this way, the *z*-approach curves are recorded with the metal in the active state (i.e., operating as a conventional amperometric SECM tip with redox mediators that are converted within the potential stability range of the elemental state of the tip metal), and then oxidized to produce the pH-sensitive metal oxide layer [78]. In other cases, positioning is achieved with dual microelectrodes with the feedback mode by adding a redox mediator to the electrolyte [45,79], a procedure that will be described in Section 3.1. Unfortunately, only a very small number of ion species can be detected using this type of microsensors (namely, $H^+$, $Ag^+$, and $Cl^-$), and indeed they have only been used as pH microscopy when characterizing thin surface layers and coatings on metals, as shown in Table 3.

**Table 3.** Metallic microsensors used as potentiometric probes in SECM characterization of thin surface layers and coatings on metals.

| Ion | Metal | Application | Reference |
|---|---|---|---|
| $H^+$ | Antimony-antimony oxide | Corrosion reactions at cut edges of galvanized steel and polymer coated galvanized steel | [42,77,80,81] |
| $H^+$ | Antimony-antimony oxide | Corrosion inhibition efficiency of 2-mercaptobenzothiazole on copper | [82] |
| $H^+$ | Antimony-antimony oxide | Corrosion inhibition efficiency of benzotriazole for the galvanic coupling of copper and iron | [83] |
| $H^+$ | Iridium-iridium oxide | Corrosion reactions at scratched alkyd-melamine coating applied on 16 MnCrS5 carbon steel | [84] |
| $H^+$ | Platinum-iridium oxide | Corrosion reactions on 316 L stainless steel surface | [79,85] |

The second type of passive potentiometric probes corresponds to ion-selective microelectrodes (ISMEs), which consist of a selective transducer (usually a membrane) that transfers the ion activity of a certain species occurring in the electrolyte phase to an electrical potential. The sensing membrane is a multicomponent solution (herein named cocktail) containing the ionophore, the polymeric matrix, the lipophilic ion exchanger, and the lipophilic salt. The ionophore is the component that selectively forms a complex with the primary ion to be monitored, whereas the polymeric matrix accounts for the mechanical stability of the system. Since membranes must be immiscible with water, lipophilic components are employed.

Although a detailed review on the use of potentiometric probes as SECM sensors can be found elsewhere [86], Table 4 lists the ISME employed to monitor thin surface layers and coatings on metals.

**Table 4.** Ion-selective microelectrodes (ISME) used as potentiometric probes in SECM characterization of thin surface layers and coatings on metals.

| Ion | Ionophore | Ion-Selective Cocktail | Application | Reference |
|---|---|---|---|---|
| $H^+$ | Hydrogen ionophore I | Cocktail B | Corrosion and self-healing functions at cut-edges of galvanized steel | [87,88] |
| $H^+$ | Hydrogen ionophore I | Cocktail B | Imaging of microdefects in sol-gel film coatings deposited on AZ31 and AZ31B magnesium alloys | [89,90] |
| $H^+$ | Hydrogen ionophore I | Cocktail B | Corrosion protection of inhibitor loaded composite coatings on AZ31 magnesium alloy | [91] |
| $H^+$ | Hydrogen ionophore II | Potassium tetrakis(4-chlorophenyl)borate and membrane solvent 2-nitrophenyloctyl ether | Inhibitor-doped hydroxyapatite (HA) microparticles applied over aluminum alloy (AA2024) | [92] |
| $H^+$ | Hydrogen ionophore I | Cocktail B | Particulate 6092-T6 Al metal matrix composites reinforced with 20 vol.% of $B_4C$, $SiC$, and $Al_2O_3$ | [93,94] |
| $H^+$ | Hydrogen ionophore I | Cocktail B | Galvanic corrosion and localized degradation of aluminium-matrix composites reinforced with silicon particulates | [95] |
| $H^+$ | Hydrogen ionophore II | Potassium tetrakis(4-chlorophenyl)borate and membrane solvent 2-nitrophenyloctyl ether | Smart coatings applied to galvanized steel | [96] |
| $H^+$ | Hydrogen ionophore II | Cocktail A | Steel samples with Al-Zn-Mg coatings | [97] |
| $H^+$ | Hydrogen ionophore I | Cocktail B | Cut edge consisting of a zinc anode and a split iron cathode | [98] |
| $Mg^{2+}$ | Magnesium ionophore II | Cocktail B | AZ31 and AZ31B magnesium alloys coated with a thin sol–gel film | [89,90] |
| $Mg^{2+}$ | Magnesium ionophore II | Cocktail B | Corrosion protection of inhibitor loaded composite coatings on AZ31 magnesium alloy | [91] |
| $Mg^{2+}$ | Bis-N,N-dicyclohexyl-malonamide | Tetrahydrofurane, poly(vinyl chloride), potassium tetrakis(4-chlorophenyl)-borate, and 2-nitrophenyl octyl ether | Galvanic corrosion of Mg coupled to Fe | [40,99,100] |
| $Mg^{2+}$ | N;N′,N″-tris[3-(heptylmethylamino)-3-oxopropionyl]-8,8′-iminodioctylamine | Potassium tetrakis(4-chlorophenyl) borate, o-nitrophenyl-n-octylether, poly(vinylchloride), and cyclohexanone | Galvanic corrosion of Mg coupled to Fe | [101] |
| $Mg^{2+}$ | Magnesium ionophore II | Cocktail A | Galvanic corrosion of Mg coupled to Fe | [102] |
| $Zn^{2+}$ | Zinc ionophore I | Tetrahydrofurane, 2-nitrophenyl octyl ether, poly(vinyl chloride), and potassium tetrakis(4-chlorophenyl)borate | Corrosion reactions at cut edges of galvanized steel and polymer coated galvanized steel | [39,42,76] |
| $Zn^{2+}$ | Zinc ionophore I | Tetra-n-butyl thiuram disulfide, sodium-tetrakis[3,5-bis(trifluoro- | Painted electrogalvanized steel with two artificial defects | [103] |

| | | | | |
|---|---|---|---|---|
| Cl⁻ | Chloride ionophores I and II | methyl)phenyl]borate, tetrakis(4-chlorophenyl)borate, and tetradodecylammonium, dissolved in 2-nitrophenyloctyl ether<br>Solvents: 2-nitrophenyl octyl ether, 2-nitrophenyl pentyl ether, 2-nitrophenyl phenyl ether, 1,2-dimethyl-3-nitrobenzene, 2-fluorophenyl 2-nitrophenyl ether, benzyl 2-nitrophenyl ether.<br>Ion exchangers: potassium tetrakis(4-chlorophenyl) borate, and tridodecylmethylammonium chloride | Cut-edge of metallic coated steel | [104] |
| Na⁺ | Sodium ionophores II, VI, VIII and X | Solvents: 2-nitrophenyl octyl ether, 2-nitrophenyl pentyl ether, 2-nitrophenyl phenyl ether, 1,2-dimethyl-3-nitrobenzene, 2-fluorophenyl 2-nitrophenyl ether, benzyl 2-nitrophenyl ether.<br>Ion exchangers: potassium tetrakis(4-chlorophenyl) borate, and tridodecylmethylammonium chloride (TDDMACl) | Cut-edge of metallic coated steel | [104] |

Chloride ionophore I (24897): meso-Tetraphenylporphyrin manganese(III)-chloride complex [$C_{44}H_{28}ClMnN_4$]. Chloride ionophore II (24901): 4,5-Dimethyl-3,6-dioctyloxy-o-phenylene-bis(mercurytrifluoroacetate) [$C_{28}H_{40}F_6Hg_2O_6$], ETH 9009. Cocktail A: 2-Nitrophenyl octyl ether, 89.3 wt.% (73732) + Sodium tetraphenylborate, 0.7 wt.% (72018). Cocktail B: 2-Nitrophenyl octyl ether, 89.3 wt.% (73732) + Potassium tetrakis(4-chlorophenyl)borate, 0.7 wt.% (60591). Hydrogen ionophore I (95292): Tridodecylamine [$CH_3(CH_2)_{11}]_3N$. Hydrogen ionophore II (95295): 4-Nonadecylpyridine ($C_{24}H_{43}N$), ETH 1907, Proton ionophore II. Magnesium ionophore II (63083): N,N''-Octamethylene-bis(N'-heptyl-N'-methyl-methylmalonamide [$C_{32}H_{62}N_4O_4$], ETH 5214. Sodium ionophore II (71733): N,N'-Dibenzyl-N,N'-diphenyl-1,2-phenylenedioxydiacetamide [$C_{36}H_{32}N_2O_4$], ETH 157. Sodium ionophore VI (71739): Bis[(12-crown-4)methyl] dodecylmethylmalonate, Dodecylmethylmalonic acid bis[(12-crown-4)methyl ester] [$C_{34}H_{62}O_{12}$]. Sodium ionophore VIII (73929): Bis[(12-crown-4)methyl] 2,2-didodecylmalonate [$C_{45}H_{84}O_{12}$]. Sodium ionophore X (71747): 4-tert-Butylcalix[4]arene-tetraacetic acid tetraethyl ester [$C_{60}H_{80}O_{12}$]. Zinc ionophore I (96491): Tetrabutylthiuram disulfide [$C_{18}H_{36}N_2S_4$].

## 3. Operation Modes

As described above, the foundation of amperometric operation is the change in the measured current ($i_T$) at the surface of a biased microelectrode occurring when it is moved near the surface of a substrate immersed in an electrolyte solution containing a redox mediator. Different operation modes can be identified depending on the origin and function of the redox mediator in the electrochemical system formed by the tip and the investigated substrate.

### 3.1. Feedback Modes

The feedback mode was one of the first operation modes employed in SECM [105], and it is one of the most frequently employed modes due to its versatility. In this mode, the tip current ($i_T$) due to the redox conversion of a redox mediator is monitored, and its magnitude varies with the tip/substrate distance ($d$), the chemical nature of the mediator as well as the composition and conductivity of the electrolytic solution. A different potential value must be applied to the tip for each redox mediator. In the presence of a non-conductive substrate, the diffusion of the mediator is hindered and eventually blocked in the proximity of the substrate. That is, the faradaic current measured at the tip, $i_T$, gradually decreases while performing an approach of the tip to the substrate because the diffusion of the mediator towards the active area of the tip is hindered by the proximity of the substrate ($i_T < i_{T,\infty}$), and this behaviour is named negative feedback (see Figure 5A). Since the underlying metal is not in direct contact with the electrolyte medium in the case of non-defective insulating coatings, the use of a redox mediator and its eventual development of a redox potential in the system produces no significant effect on the investigated system. Although the information provided by the technique has only spatial resolution (i.e., topography and morphology), this mode has found application for the investigation of transport phenomena through defect-less barrier organic coatings applied on metallic substrates leading to mechanistic information on water uptake [106–108] and lixiviation processes [109], as well as the detection of the early stages of coating blistering and delamination induced by ionic species such as chloride [110–113].

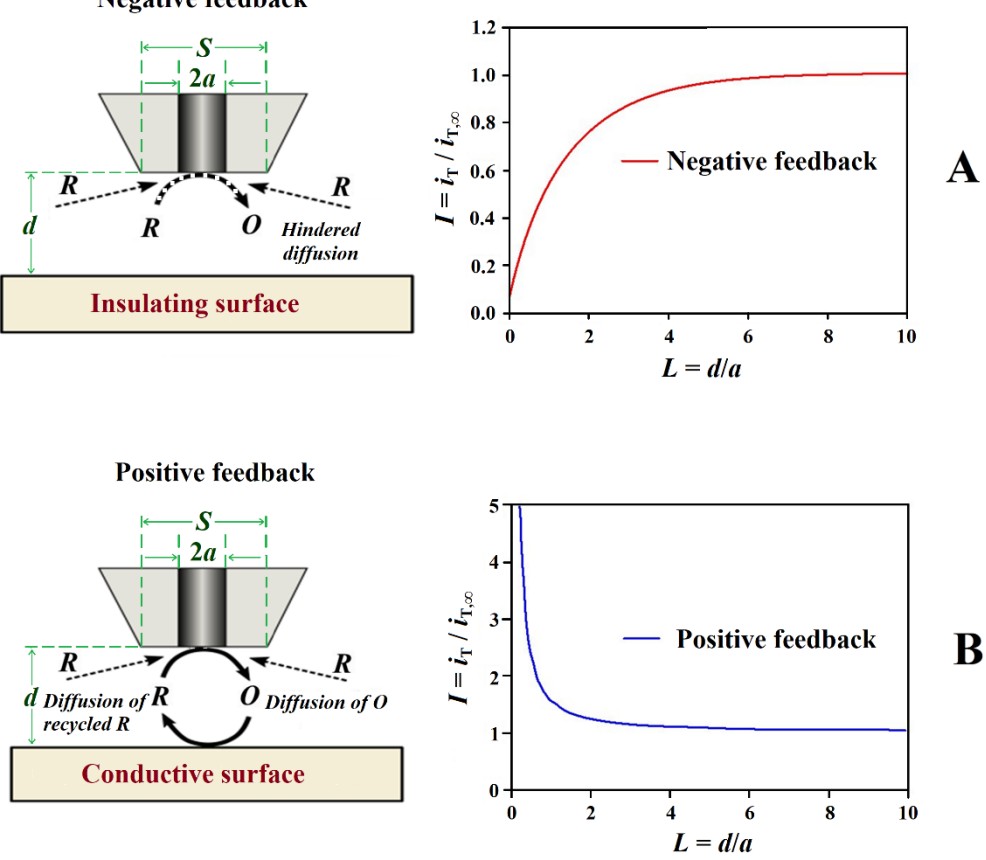

**Figure 5.** Schemes and shapes of Z-approach curves in the feedback mode of SECM [114]. Types of feedback: (**A**) negative and (**B**) positive.

Conversely, if the surface of the sample is conductive, the mediator can be regenerated on it, and an increase of $i_T$ can be observed ($i_T > i_{T,\infty}$) for smaller tip-substrate distances

originating a positive feedback behaviour (cf. Figure 5B). The study of electrically insulating or conductive surfaces is possible thanks to the appearance of negative or positive feedback effects, obtaining images of the studied surface that reflect the occurrence of defects in insulating coatings, including both inorganic and organic matrices, to be investigated [41,115–117]. Depending on the size of the tip, the measurement of $i_T$ can thus provide information about sample topography and its electrical and chemical properties, allowing for the occurrence of defects ranging from pinholes to holidays and scratches, to be detected, as well as to monitor their evolution [115–117]. In brief, insulating regions produce changes in the current measured at the tip due to topographic changes that modify the transport regime of the redox mediator from the electrolyte bulk towards the tip, frequently interfering with the signal while scanning the substrate in close proximity. Conversely, regions in the substrate that are conductive and capable of regenerating the redox mediator produce an increase in current measured at the tip. It should be noted that, since the current response in feedback mode is highly dependent on tip-to-substrate distance, it is preferable to use as small a distance as possible (without crashing the tip) to increase sensitivity.

In the case of the feedback operation mode, kinetic information can also be extracted from the experimental approach curves after taking in account the geometric factors of the tip [118], which is of direct application for the determination of the rate constants associated with the formation of corrosion inhibitor layers on metals and their ageing by thickening or greater compactness [119–123]. Additionally, the method has also been employed to gain information on the adsorption of the inhibitor molecules on the metal [54].

A total of 18 different mediators have been employed until now in feedback operation, as listed in Table 5. The most used is ferrocene-methanol (FcMeOH) (ca. 36% of the cases) followed by hexacyanoferrate (III), $[Fe(CN)_6]^{3+}$, (with almost 17% of the cases). This fact is all the more striking as a wide variety of substrates are involved in the studies, and their characteristic potential values in the test environment vary considerably from highly reactive metals such as magnesium alloys to more noble materials such as copper. That is, it is often not taken into account that the coexistence of the two forms of a redox mediator confers a potential on the sample that effectively acts as if an external bias were applied, so that the same mediator should not be the preferred choice for widely dissimilar metals.

**Table 5.** Redox mediators used in SECM feedback mode for the characterization of thin surface layers and coatings on metals.

| Mediator | Electrolyte Solution | $E_{tip}$ | Substrate | Coating | Tip ($\varphi$, μm) | Reference |
|---|---|---|---|---|---|---|
| AB | 1 mM AB + 0.1 M Bu₄NBF₄ in 0.1 M ACN | −1.6 V vs. SCE | Glass | Poly(3,4-ethylenedioxythiophene) (PEDOT) | Pt (10) | [124] |
| H⁺ | 1 mM BQ + 0.1 M Bu₄NBF₄ in 0.1 M ACN | −0.5 V vs. SCE | Glass | PEDOT | Pt (10) | [124] |
| H⁺ | 0–50 mM p-BQ M in PC ethyleneglycol + 10–50 mM p-BQ + 20 mM KI | −1.0 V vs. NHE | Glass | AgCl-coated sample | Pt (10) | [125] |
| H⁺ | | | Teflon | | | |
| DcMeFc | 0.5 mM DcMeFc + 5 mM BATB in DCE | – | Pt bands over glass | Parylene C | Pt_pC (25) | [47] |
| DMAMFc | 1.25 mM DMAMFc⁺ + 10 mM H₂SO₄ | +0.40 V vs. Ag/AgCl | AA2024-T3 aluminum alloy | Poly(aniline) + poly-(methylmethacrylate) (PANI-PMMA) blend | Pt (10) | [126] |
| | 1.25 mM DMAMFc + 10 mM BBS | +0.80 V vs. Ag/AgCl | AA2024-T3 aluminum alloy | Zr(IV)–alkyl-phosphonate | Pt (10) | [127] |

| | | | | | | |
|---|---|---|---|---|---|---|
| | | +0.16 V vs. SCE | | Zr(IV)–aryl-phosphonate | | |
| Fe | 0.01 M $Fe^{3+}$ + 0.5 M $H_2SO_4$ | −0.1 V vs. Ag/AgCl | – | Polyester-polypyrrole-graphene oxide (PPy/GO) | Pt (100) | [51] |
| | 1 mM of Fc + 0.1 M $Bu_4NBF_4$ in ACN | +0.4 V vs. SCE | Glass | PEDOT | Pt (10) | [124] |
| Fc | 0.1 M $Bu_4NBF_4$ + 0.1 M in ACN | +0.41 V vs. Ag/AgCl | Glass | Zn-porphyrin layers on indium tin oxide (ITO) electrode and ITO modified with poly-1 | Pt (10) | [128] |
| FcMeOH | 0.5 mM FcMeO + 0.1 M KCl | +0.5 V vs. Ag/AgCl/KCl sat. | Carbon steel (CS) | Two-component polyurethane | Pt (10) | [66] |
| FcMeOH | 0.5 mM FcMeOH + 0.1 M NaCl | +0.8 V vs. Ag/AgCl | AA2024 aluminum alloy | 1% γ-aminopropyltrimethoxy silane (γ-APS)-doped and 2.5% bis-1,2-[triethoxysilyl]ethane (BTSE)-doped epoxy coating | Pt (25) | [129] |
| FcMeOH | 0.9 mM FcMeOH + 5 wt.% NaCl | – | Q235 mild steel (MS) | Enamel coating | – | [130] |
| FcMeOH | 0.5 mM FcMeOH + 3.5 wt.% NaCl | +0.5 V vs. Ag/AgCl/KCl sat. | High strength steel (SAPH440) | Enamel coating | – | [131] |
| FcMeOH | 5 mM FcMeOH + 0.05 M NaCl | +0.6 V vs. Ag/AgCl/KCl sat. | AA2024-T3 aluminum alloy | Epoxy coating and epoxy coating containing silyl-ester doped capsules | Pt (5) | [132] |
| FcMeOH | 5 mM FcMeOH + 0.5 mM NaCl | +0.5 V vs. Ag/AgCl/KCl sat. | Coil coated steel (CCS) | Polyester paint | Pt (10) | [106] |
| FcMeOH | 0.1 M TBATFB in PC or ACN | +0.4 V vs. Ag/AgCl | Gold-coated silicon and p-Si | Oligothiophenes | Pt (25) | [133] |
| FcMeOH | 1 mM FcMeOH + 0.1 M $KNO_3$ | +0.35 V vs. Ag/AgCl | 2024-T3 aluminum alloy | Non-chromated primer on anodized Al; waterborne primer on alodine pretreated Al; chromated primer on alodine pretreated Al | Pt (25) | [134] |
| FcMeOH | 0.9 mM FcMeOH + 0.1 M $KNO_3$ | +0.4 V vs. Ag/AgCl | Silicon | $Cu_xS$ substrate + $SiO_2$/Si | Pt (10) | [135] |
| FcMeOH | 10 mM FcMeOH + 0.1 M TBAPF6 in DMF | – | Pt | PEDOT | Pt (30) | [136] |
| FcMeOH | 2.2 mM FcMeOH + 0.1 M $KNO_3$ | +0.2 V vs. Ag-QRE | Pt bands over glass | Parylene C | $Pt_{pC}$ (25) | [47] |
| FcMeOH | 0.5 mM FcMeOH + 0.05 M NaCl | +0.4 V vs. Pt-wire | Nickel foil | Plasticized Polyvinyl chloride (PVC) | Pt (10) | [109] |
| FcMeOH | 1 mM FcMeOH + 0.1 M $KNO_3$ | +0.4 V vs. Ag/AgCl | Cu | Monolayer of $C_{12}H_{25}$–X (X = –SH, –S–S–, –SeH and –Se–Se–) | Pt (10) | [137] |

| | | | | | | |
|---|---|---|---|---|---|---|
| FcMeOH | 1 mM FcMeOH + 0.1 M KNO$_3$ or K$_2$SO$_4$ | +0.4 V vs. Ag/AgCl | Copper-based quaternary bronze (UNS C83600) | Cu Patina | Pt (25) | [138] |
| FcMeOH | 1 mM FcMeOH + 0.1 M KCl | +0.4 V vs. Ag/AgCl | – | Polytetrafluorethylene (PTFE) | BDD (6 to 23) | [53] |
| FcMeOH | 1 mM FcMeOH + 1 mM Na$_2$SO$_4$ | +0.50 V vs. Ag/AgCl/KCl (3 M) | Cu | 2-Mercaptobenzimidazole (2-MBI) | Pt (10) | [123] |
| FcMeOH | Ringer's physiological solution | +0.47 V vs. SCE | Ti-6Al-4V and Ti-21Nb-15Ta-6Zr alloys | HA–ZrO$_2$ | Pt (12.5) | [116] |
| FcMeOH | 0.67 mM FcMeOH + 0.067 M Na$_2$SO$_4$ + 0.33 mM BTAH | +0.50V vs. Ag/AgCl/KCl (3 M) | Cu | Benzotriazole (BTAH) | Pt (25) | [122] |
| FcMeOH | 1 mM FcMeOH + 0.2 M KCl | +0.4 V vs. Ag/AgCl | AA2024-T3 aluminum alloy | Epoxy resin + vanadate- and tungstate-doped PPy/Al flake composite pigments | Pt (10) | [139] |
| FcMeOH | 1 mM FcMeOH + 0.1 M K$_2$SO$_4$ | +0.6 V vs. Ag/AgCl | Inconel 625 | Thin coatings of the alloy on MS using High velocity oxy-fuel (HVOF) | Pt (4) | [140] |
| FcMeOH | 5 mM FcMeOH + 0.01 M NaCl | +0.5 V vs. Ag/AgCl/KCl sat. | 2024 aluminium alloy | Acrylic coat (undoped coating system) and with mesoporous pretreatment | Pt (10) | [141] |
| FcMeOH | 0.5 mM FcMeOH + 0.1 M KCl | +0.5 V vs. Ag/AgCl, KCl sat. | MS | Polyester | Pt (10) | [13] |
| FcMeOH | 0.5 mM FcMeOH + 0.1 M KCl | +0.50 V vs. Ag/AgCl, KCl sat. | MS | Polyester | Pt (10) | [142] |
| FcMeOH | 0.5 mM FcMeOH + 0.1 M KCl and 0.5 mM FcMeOH + 0.1 M K$_2$SO$_4$ | +0.5 V vs. Ag/AgCl | MS | Polyurethane | Pt (10) | [111] |
| FcMeOH | 1 mM FcMeOH + 0.1 M Na$_2$SO$_4$ | – | MS | Inconel 625 formed using a HVOF | Pt (10) | [143] |
| FcMeOH | 1 mM FcMeOH + 0.1 M NaCl | – | Q235 MS | Graphene oxide–mesoporous silicon dioxide layer–nanosphere structure loaded with tannic acid (GSLNTA) | Pt (25) | [36] |
| FcMeOH | 1 mM FcMeOH + 0.1 M KCl | – | 5083 aluminum alloy | Hexamethylene diisocyanate trimer (HDIt) microcapsules into epoxy | Pt (25) | [144] |
| FcMeOH | 0.9 mM FcMeOH + 0.5 M NaCl | + 0.5 V Ag/AgCl | CS | Organosol | Pt (10) | [145] |
| FcMeOH | 0.5 mM FcMeOH + 0.1 M or 0.62 M NaCl | +0.45 V vs. Ag/AgCl/KCl (3 M) | AZ91D magnesium alloy | Surface layer formed by Micro-arc oxidation (MAO) | Pt (10) | [146] |
| FcMeOH | 1 mM FcMeOH + 100 mM Na$_2$SO$_4$ | +0.45 V vs. SCE | Cu | BTAH | Pt (25) | [119] |

| | | | | | | |
|---|---|---|---|---|---|---|
| | and 0.667 mM FcMeOH + 67 mM Na$_2$SO$_4$ + 0.333 mM BTAH | | | | | |
| FcMeOH | 2 mM FcMeOH + 0.2 M Na$_2$SO$_4$ with 0.2% (v/v) ethanol | – | Cu | Poly(3-ethoxy-thiophene) (PEOT) and poly(ethylenedioxy-thiophene) (PEDT) | Pt (10 and 25) | [147] |
| FcMeOH | 1 mM FcMeOH + 0.1 M NaCl | +0.4 V vs. SCE | Cu | Self-assembled monolayers (SAMs) formed by HL | Pt (25) | [148] |
| FcMeOH | 0.5 mM FcMeOH + 0.1 M KCl or 0.1 M K$_2$SO$_4$ | +0.5 V vs. Ag/AgCl/KCl sat. | CCS | Polyester (PES) | Pt (10) | [107] |
| FcMeOH | 0.5 mM FcMeOH + 0.1 M PBS | 0.6 V vs. Ag/AgCl | MS | Electrodeposited silica | Pt (10) | [149] |
| [Fe(CN)$_6$]$^{3-}$ | 1 mM K$_4$[Fe(CN)$_6$] | +0.4 V vs. Ag/AgCl/KCl sat. | CS | Two-component polyurethane | Pt (10) | [66] |
| [Fe(CN)$_6$]$^{3-}$ | 0.1 M KCl | +0.45 V vs. Ag/AgCl/KCl sat. | Pt and SS | Amorphous alumina thin film grown by Metal organic chemical vapour deposition (MOCVD) process | Pt (25) | [115] |
| [Fe(CN)$_6$]$^{3-}$ | 0.5 mM Ferrocyanide + 44 mM PBS | +0.5 V vs. Ag/AgCl/KCl (3 M) | Gold disk electrode | Cytochrome C | Pt (25) | [150] |
| [Fe(CN)$_6$]$^{3-}$ | 10 mM K$_3$Fe(CN)$_6$ + 0.1 M Na$_2$SO$_4$ | −0.4 V vs. SCE | Pt | Nafion film containing Os(pbsy) | Pt (10.1) | [151] |
| [Fe(CN)$_6$]$^{3-}$ | 1 mM K$_4$[Fe(CN)$_6$] + 0.1 M NaCl | +0.40 V vs. Ag/AgCl, KCl sat. | GS | Two-component epoxy primer containing zinc phosphate | Pt (10) | [152] |
| [Fe(CN)$_6$]$^{3-}$ | 50 mM K$_3$Fe(CN)$_6$ + 1 M Na$_2$SO$_4$ | −0.6 V vs. Ag/AgCl | Pt/Glass | PVC over Pt sheet; poly-terthiophene on glass | Au (5 and 25) | [52] |
| [Fe(CN)$_6$]$^{3-}$ | 10 mM K$_4$[Fe(CN)]$_6$ + 3 wt.% NaCl 3% | – | Steel | Epoxy resins (Diglycidylether of Bisphenol A (DGEBA) + Methylpentanediamine (DAMP)) | – | [153] |
| [Fe(CN)$_6$]$^{3-}$ | 2 mM Fe(CN)$_6$$^{3-}$/Fe(CN)$_6$$^{4-}$ + 0.1 M KCl | +0.5 V vs. Ag/AgCl/KCl sat. | Au | Thioglycolic acid (TGA) + Quercetin (Q) | Pt (10) | [154] |
| [Fe(CN)$_6$]$^{3-}$ | 5 mM K$_4$[Fe(CN)]$_6$ + 0.1 M KBr | +0.5 V vs. Ag/AgCl | Pt | Polytetrafluoroethylene (PTFE) plaques | Pt (50) | [49] |
| [Fe(CN)$_6$]$^{3-}$ | 4 mM K$_4$[Fe(CN)]$_6$ + 0.1 M KNO$_3$ | – | Pt | Pt/400-nm thick layer of a-Si:H Langmuir-Blodgett films of iron oxides nanoparticles | Pt (5) | [155] |
| [Fe(CN)$_6$]$^{3-}$ | 5 mM Fe(CN)$_6$$^{3-}$ + 0.1 M KCl | – | – | Vinylic monomers | Pt (25 and 100) | [50] |

| | | | | | | |
|---|---|---|---|---|---|---|
| $[Fe(CN)_6]^{3-}$ | 0.1 M KCl | +0.5 V vs. Ag/AgCl | Glassy carbon substrate electrodes (GCEs) | Bismuth film | Pt (25) | [156] |
| $[Fe(CN)_6]^{3-}$ | 5 mM $K_4[Fe(CN)]_6 \cdot 3H_2O$ + 0.1 M KCl | +0.5 V vs. Ag/AgCl | Low CS Q-Panel S | Epoxy resin with Tetraethoxysilane (TEOS) and epoxy resin filled GO | Pt (10) | [157] |
| $[Fe(CN)_6]^{3-}$ | 1 mM $K_4[Fe(CN)]_6$ + 0.1 M KPF6 | +0.5 V vs. Ag/AgCl | Gold | Azido-terminated self-assembled monolayers | Pt (10) | [158] |
| $[Fe(CN)_6]^{3-}$ | 5 mM $K_3[Fe(CN)]_6$ + 100 mM KCl | −0.25 V vs. Ag/AgCl/KCl sat. | Steel | Physical vapour deposition (PVD) TiN coatings | Pt (15) | [159] |
| $[Fe(CN)_6]^{3-}$ | 10 mM $K_4[Fe(CN)]_6$ + 3 wt.% NaCl | – | Steel | Epoxy resin with and without $TiO_2$ | Pt (10) | [112] |
| $[Fe(CN)_6]^{3-}$ | 10 mM $Ru(NH_3)_6Cl_3$, $K_3Fe(CN)_6$ and $K_4Fe(CN)_6$ + 0.1 M KCl | +0.4 V vs. Ag/AgCl/KCl (3 M) | – | Polyester coated with reduced GO | Pt (100) | [160] |
| $[Fe(CN)_6]^{3-}$ | 0.5 mM $K_4[Fe(CN)_6]$ + 3.5% NaCl | +0.40 V vs. Ag/AgCl/KCl (1 M) | CS | Epoxy with zinc phosphate pigment | Pt (10) | [161] |
| $H_2$ | 0.5 mM $HClO_4$ and 0.3 mM $HClO_4$ + 0.1 M $LiClO_4$ | −0.65 V vs. Ag/AgCl | Stainless steel (SS) | Pt-$TiO_2$ prepared by MOCVD | Pt (25) | [162] |
| $H_2$ | 0.1 M HCl | 0.0 | Q235 CS | Epoxy coating sample containing zeolitic imidazole framework (ZIF-7) | – | [163] |
| $I^-$, $I_2$ | NaCl + KI (no concentrations specified) | – | X80 pipeline steel | Epoxy resin E51 and polyether amine D230 | Pt (10) | [164] |
| $IrCl_6$ | 1 mM $IrCl_6^{2-}$ + 0.1 M KCl | – | – | PTFE | BDD (6 to 23) | [53] |
| MV | 5 mM MV + 0.1 M KCl | −0.9 V vs. Ag/AgCl | Glass | Composite silica glass containing copper salts | Pt (10) | [165] |
| $O_2$ | 0.1 M KCl | −0.70 V vs. Ag/AgCl/KCl sat. | CS | Two-component polyurethane film | Pt (10) | [66,166] |
| $O_2$ | 2.2 mM FcMeOH + 0.1 M KNO3 | −0.8 V vs. Ag-QRE | Pt bands over glass | Parylene C | $Pt_{pC}$ (25) | [47] |
| $O_2$ | PBS pH 7.1 + 0.1 M NaCl + 0.01 M $NaH_2PO_4$ | −0.75 V vs. Ag/AgCl, KCl sat. | FTO | Octadeciltrichlorosilane (OTS) based SAMs | Pt (20) | [167] |
| $O_2$ | 0.1 M KF | −0.70 V vs. Ag/AgCl | magnesium mechanically reinforced by powder metallurgy Mg(PM) | Fluorine conversion coatings | Pt (10) | [168] |
| $O_2$ | 3.5 wt.% NaCl | −0.70 V vs. Ag/AgCl/KCl sat. | CS | Epoxy-$ZrO_2$ | Pt (10) | [169] |

| | | | | | | |
|---|---|---|---|---|---|---|
| $O_2$ | 0.1 M KCl | −0.70 V vs. Ag/AgCl/KCl sat. | MS | Polyurethane | Pt (10) | [170] |
| $O_2$ | 0.5 mM FcMeOH + 0.1 M KCl | +0.50 V vs. Ag/AgCl/KCl sat. | MS | Polyester | Pt (10) | [142] |
| $O_2$ | 0.1 M KCl + 0.5 mM FcMeOH and 0.1 M $K_2SO_4$ + 0.5 mM FcMeOH | −0.6 V vs. Ag/AgCl | MS | Polyurethane | Pt (10) | [111] |
| $O_2$ | 3.5 wt.% NaCl | −0.7 V vs. SCE | CS | Styrene-acrylic + terpolymer | – | [171] |
| $O_2$ | 3.5 wt.% NaCl | −0.7 V vs. Ag/AgCl/KCl sat. | AA7075 | Berberine | Pt (10) | [172] |
| 4 NB | 0.1 M $NBu_4PF_6$ in ACN | −0.60 V vs. Ag/AgCl | Glass | Zn-porphyrin layers on ITO electrode and ITO modified with the poly-1 | Pt (10) | [128] |
| TMPD | 0.1 M (TBA)$BF_4$ + 0.76 mM TMPD in ACN | – | Au | Fullerene | Pt (25) | [173] |
| TCNQ | 1 mM TCNQ + 0.1 M $NBu_4BF_6$ in ACN | +0.1 V vs. SCE | Glass | PEDOT | Pt (10) | [124] |
| TCNQ | 0.1 M $NBu_4PF_6$ in ACN | +0.27 V vs. Ag/AgCl | Glass | Zn-porphyrin layers on ITO electrode and ITO modified with the poly-1 | Pt (10) | [128] |
| TTF | 0.1 M $NBu_4PF_6$ in ACN | +0.35 V vs. Ag/AgCl | Glass | Zn-porphyrin layers on ITO electrode and ITO modified with the poly-1 | Pt (10) | [128] |
| $[Ru(NH_3)_6]^{3+}$ | 1 mM $Ru(NH_3)_6Cl_3$ + 0.1 M KCl | −0.35 V vs. Ag/AgCl/KCl sat. | Pt and SS | Amorphous alumina thin film by the MOCVD process | Pt (25) | [115] |
| $[Ru(NH_3)_6]^{3+}$ | 1 mM $Ru(NH_3)_6Cl_3$ + 0.1 M $Na_2SO_4$ | – | Silicon wafers | Pt/$Al_2O_3$ samples | Pt (25) | [174] |
| $[Ru(NH_3)_6]^{3+}$ | 1 mM $Ru(NH_3)_6Cl_3$ + 0.1 M $Na_2SO_4$ | – | SS | Pt-$TiO_2$ prepared by the MOCVD procedure | Pt (25) | [162] |
| $[Ru(NH_3)_6]^{3+}$ | 1 mM $Ru(NH_3)_6Cl_3$ + 0.1 M KCl | −0.35 V vs. Ag/AgCl/KCl sat. | Pt wire | Pt/$Al_2O_3$ | Pt (5 to 25) | [175] |
| $[Ru(NH_3)_6]^{3+}$ | 1 mM $Ru(NH_3)_6Cl_3$ + 0.1 M KCl | +0.2 V vs. Ag/AgCl | Steel | Polyester paint | Pt (25) | [176] |
| $[Ru(NH_3)_6]^{3+}$ | 0.1 M KCl | −0.4 V vs. Ag/AgCl | GCEs | Bismuth film | Pt (25) | [156] |
| $[Ru(NH_3)_6]^{3+}$ | 1 mM $Ru(NH_3)_6^{3+}$ + 0.1 M KCl | – | – | PTFE | BDD (6 to 23) | [53] |
| $[Ru(NH_3)_6]^{3+}$ | 1 mM $Ru(NH_3)_6Cl_3$ + 0.1 M KCl | −0.4 mV vs. Ag/AgCl | Glass | PES and PES-PPy/$PW12O40$ | Pt (100) | [177] |
| $[Ru(NH_3)_6]^{3+}$ | 5 mM $Ru(NH_3)_6Cl_3$ + 0.1 M KCl $K_2SO_4$ | −0.70 V vs. Ag/AgCl | NiTi | Electrodeposited tantalum layer | Pt (10) | [178] |
| $[Ru(NH_3)_6]^{3+}$ | 0.01 M $Ru(NH_3)_6Cl_3$ + 0.1 M $Na_2SO_4$ or NaCl | −0.3 V vs. Ag/AgCl | PES, PES–PANI/$HSO_4^-$ and PES–PANI/$Cl^-$ | Pt (100) | [179] |
| $[Ru(NH_3)_6]^{3+}$ | 0.01 M $Ru(NH_3)_6Cl_3$ + 0.1 M KCl | −0.4 V vs. Ag/AgCl | PES and PES-PPy/AQSA | PES and PES-PPy/AQSA | Pt (100) | [180] |

| | | | | | | |
|---|---|---|---|---|---|---|
| $[Ru(NH_3)_6]^{3+}$ | 0.01 M [Ru(NH$_3$)$_6$Cl$_3$, K$_3$Fe(CN)$_6$ and K$_4$Fe(CN)$_6$] in 0.1 M KCl | −0.4 V vs. Ag/AgCl/KCl (3 M) | – | Polyester fabrics coated with Reduced graphene oxide (RGO) | Pt (100) | [160] |
| $[Ru(NH_3)_6]^{3+}$ | 0.01 M Ru(NH$_3$)$_6$Cl$_3$ + 0.1 M KCl | −0.4 V vs. Ag/AgCl | – | PES, PES-PPy/PW$_{12}$O$_{40}$$^{3-}$ and PES-PPy/PW$_{12}$O$_{40}$$^{3-}$ + PANI | Pt (100) | [181] |
| $[Ru(NH_3)_6]^{3+}$ | 0.01 M Ru(NH$_3$)$_6$Cl$_3$ + 0.1 M KCl | −0.4 V v. Ag/AgCl | – | RGO | Pt (100) | [182] |
| $[Ru(NH_3)_6]^{3+}$ | 0.01 M Ru(NH$_3$)$_6$Cl$_3$ + 0.1 M KCl | −0.4 V vs. Ag/AgCl/KCl (3 M) | – | PES-PPy/GO (10%, 20% and 30%) | Pt (100) | [51] |
| $[Ru(NH_3)_6]^{3+}$ | 01 mM Ru(NH$_3$)$_6$Cl$_3$ + 0.1 M KCl | −0.35 V vs. Ag/AgCl | Pt | Alumina | Pt (25) | [183] |

Since some of these mediators were monitored at specific redox potential values (e.g., O$_2$ or iodine), Table 5 includes that information together with the tip composition and size, whereas the applications are described in terms of the coating or surface film, test environment, and substrate compositions. Often SECM measurements are carried out in aerated chloride-containing electrolytes as a supporting solution due to their aggressive nature toward metals, typically in concentrations close to 3.5 wt.% or 0.1 M. However, simulated biological fluids (SBF) like Ringer's solution have been employed in experiments involving alloys used in biomedical implants [116].

### 3.2. Generation-Collection Modes

The term generation/collection mode encompasses two different modes of amperometric operations in SECM: tip generation/substrate collection (TG/SC) (see Figure 6A,B), and substrate generation/tip collection (SG/TC) (in Figure 6C), the main difference being the site at which the redox reaction employed for imaging occurs; either at the substrate or at the tip [20]. Although the tip and the substrate both act as working electrodes, the corrosion processes at the substrate are sufficient to develop a spontaneous potential that sustains the reaction without the need to polarize the substrate [184]. In this case, only the application of a potential to the SECM tip is necessary to measure the current flowing at the tip. An alternative situation occurs when the bipotentiostat is employed to set the substrate potential as well as the tip, since the instrument can be used to measure current in both the SECM tip and the substrate.

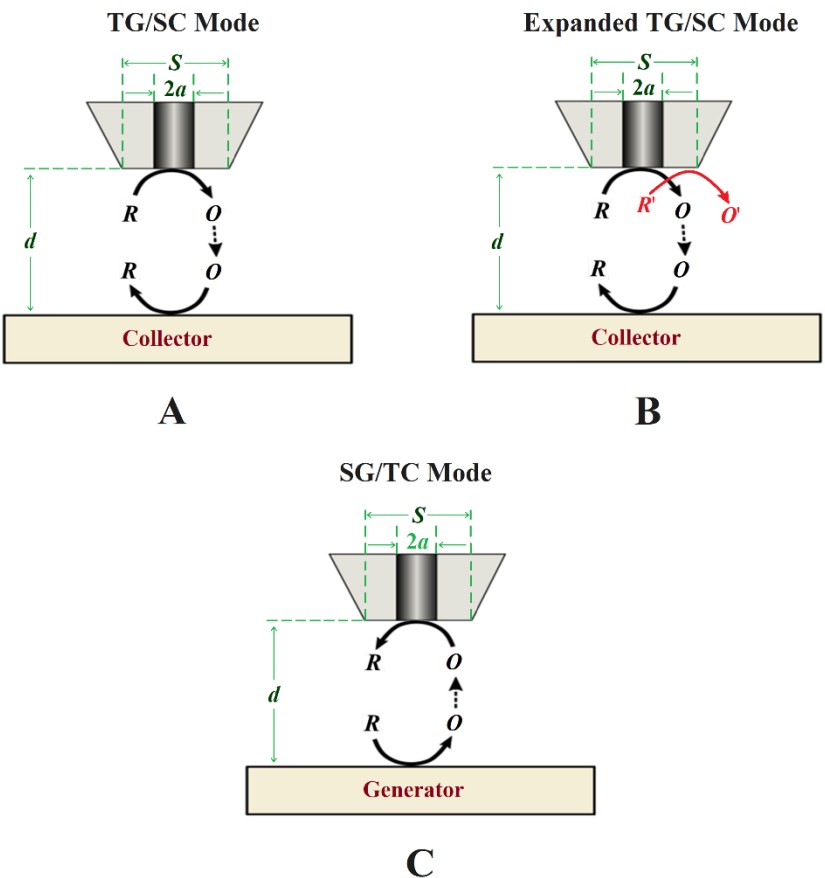

**Figure 6.** Schemes of generation-collection (G/C) modes of SECM. Types of G/C: (**A**) conventional tip generation and substrate collection, (**B**) bi-reaction tip generation and substrate collection, and (**C**) sample generation and tip collection.

In the TG/SC mode, the electroactive species that can be detected at the substrate is generated at the tip. In this case, the tip and the substrate must have different potentials, either using the bipotentiostat or by the substrate developing a different open circuit potential (OCP). A multireaction TG/SC mode was introduced by Leonard and Bard [185] in 2013 by using the redox conversion of two different species at the tip, as shown in Figure 6B. Depending on the potential required to reduce $O$ to $R$ or $O'$ to $R'$ at the substrate, if it is different for both reactions and a total collection efficiency of $O$ from $R$ can be set at the tip, then the current associated with each of the two reduction reactions can be separated. In the specific case of corrosion systems, other possibilities for bi-reaction interfaces can include two oxidation reactions or a reduction combined with an oxidation.

In SG/TC, the current at the microelectrode arises from a species generated at the surface of the substrate (Figure 6C). This is the traditional G/C mode, and it has an important application for determining the reaction rates in function of the tip-to-substrate distance. If $R$ reacts during its transport from the tip to the substrate, the relation between the current intensity at the substrate and at the tip becomes smaller and will greatly vary with the distance, $d$, and it can be used to obtain the rate constant of the homogeneous reaction.

In the G/C mode, the most frequently encountered situation is that redox mediators are selected among the species generated at the substrate under study under active corrosion conditions. The detection of the chemical species involved in the metallic corrosion process allows obtaining concentration profiles in the adjacent electrolyte to the sample. According to Table 6, five mediators have been employed, with the Fe ion as the most used (62% of the cases), followed by the hydrogen ion.

**Table 6.** Redox mediators used in generation-collection mode SECM for the characterization of thin surface layers and coatings on metals.

| Mediator | Electrolyte Solution | $E_{tip}$ | Substrate | Coating | Tip ($\varphi$, μm) | Reference |
|---|---|---|---|---|---|---|
| Fe | 5–7 mM FeSO$_4$(NH$_4$)$_2$SO$_4$ + 0.1 M KCl | +0.77 V and +0.60 V vs. Ag/AgCl/KCl sat. | CS | Two-component polyurethane | Pt (10) | [66,166] |
| Fe | 5 mM FcMeOH + 0.05 M | +0.6 V vs. Ag/AgCl/KCl sat. | AA2024-T3 | Epoxy coating and epoxy coating containing silylester doped capsules | Pt (5) | [132] |
| Fe | 3.5 wt.% NaCl | +0.60 V vs. Ag/AgCl | CS | Epoxy | Pt (10) | [186] |
| Fe | 0.1 M NaCl | +0.60 V vs. Ag/AgCl, KCl sat. | HT steel | Epoxy | Pt (10) | [187] |
| Fe | 1 M NaClO$_4$ and 1 mM HClO$_4$ | +0.5 V vs. Ag/AgCl/KCl (3 M) | Steel | Two-component epoxy | Pt (10) | [188] |
| Fe | 0.1 M KCl | +0.6 V vs. Ag/AgCl/KCl sat. | MS | Polyurethane | Pt (10) | [170] |
| Fe | 0.1 M NaCl | +0.6 V vs. Ag/AgCl/KCl sat. | Steel | CrN and TiN PVD | Pt (10) | [189] |
| Fe | 3.5 wt.% NaCl | +0.6 V vs. Ag/AgCl | MS | Epoxy + WO$_3$ nanoparticle | Pt (10) | [190] |
| Fe | Natural seawater | +0.3 V vs. Ag/AgCl | MS | DGEBA + CeO$_2$ nanoparticles | Pt (10) | [191] |
| Fe | Natural seawater | +0.60 V vs. Ag/AgCl | MS | Neat epoxy and epoxy-(3-aminopropyl)triethoxysilane (APTES) modified MoO$_3$ nanocomposite | Pt (10) | [192] |
| Fe | 0.1 M NaCl | +0.60 V vs. Ag/AgCl/KCl sat. | CS | Epoxy coatings containing magnesium nanoparticles | Pt (10) | [193] |
| Fe | 0.1 M NaCl | +0.60 V v Ag/AgCl/KCl sat. | Mn steel | Epoxy | Pt (10) | [194] |
| H$_2$O$_2$ | 0.1 M KCl | – | MS | Polyurethane | Pt (10) | [170] |
| H$_2$O$_2$ | 0.1 M KCl | +0.25 V vs. Ag/AgCl/KCl sat. | CS | Polyurethane | Pt (10) | [66] |
| O$_2$ | 0.1 M KCl | +0.4 V and +0.7 V vs. Ag/AgCl/KCl sat. | CS | Two-component polyurethane | Pt (10) | [166] |
| H$_2$ | SBF | 0.0 V vs. Ag/AgCl/KCl (3 M) | AZNd Mg alloy | Mg(OH)$_2$ passive layer | Pt (25) | [195] |

| H₂ | 1 mM Pr(NO₃)₃ + SBF | 0.0 V vs. Ag/AgCl | AZNd Mg alloy | Praseodymium conversion layers | Pt (25) | [196] |
|---|---|---|---|---|---|---|
| H₂ | 0.01 M NaCl | −0.05 V vs. Ag/AgCl-QRE | AZ31B magnesium alloy | PEDOT | Pt/IrOx (25) | [48] |
| H₂ | SBF | 0.0 V vs. Ag/AgCl | AZ31 magnesium alloy | Coating induced by phosphate-based ionic liquids | Pt (10) | [197] |
| Ru(NH₃)₆]³⁺ | 5 mM Ru(NH₃)₆Cl₃ + 0.1 M K₂SO₄ | +0.1 V vs. Ag/AgCl | NiTi | Electrodeposited tantalum layer | Pt (10) | [178] |

*3.3. Redox Competition Mode*

The redox competition mode was introduced by Schuhmann and co-workers in 2006 [198] in a work related to catalysis. In this mode, the SECM tip and the substrate are polarized using the bipotentiostat. When the tip and the substrate are close to each other, they compete for the same redox species (see Figure 7), although the current is measured only at the tip. In a typical corrosion system formed by a metal covered by a non-conductive coating, the oxygen reduction current measured at the SECM tip remains constant while the tip is scanned over the non-defective coating [199]. However, if a scratch is made through the coating system and the metal is exposed to the solution, the current measured at the SECM tip will decrease as the tip explores this active area because the redox species (e.g., oxygen) is consumed at both working electrodes [152,200], and the decrease observed can be correlated to the chemical activity of the substrate that corrodes after its direct exposure to the aggressive environment.

## Redox Competition Mode

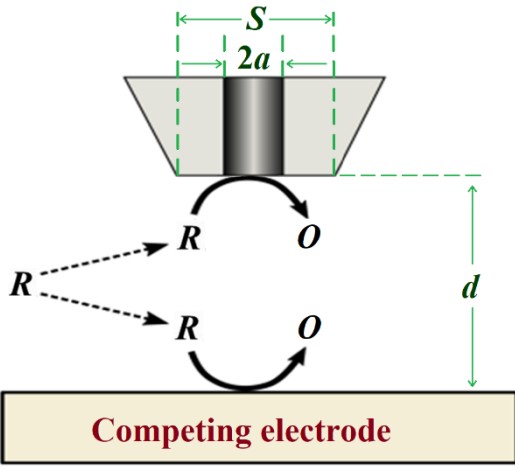

**Figure 7.** Scheme of the redox competition mode of SECM.

In this mode, dissolved molecular O₂ has been used as a redox mediator in all cases, as shown in Table 7. The most usual application consists in producing a scratch to the coating in order to allow the exposure of the underlying metal to the environment. With this operation mode and using O₂ as mediator, the competition between the O₂ consumed at the tip and that at the substrate (related to the corrosion reaction) can be monitored, even for non biased substrates [1].

**Table 7.** Redox mediators used in redox competition mode SECM for the characterization of thin surface layers and coatings on metals.

| Mediator | Electrolyte Solution | $E_{tip}$ | Substrate | Coating | Tip ($\varphi$, μm) | Reference |
|---|---|---|---|---|---|---|
| $O_2$ | 0.5 mM FcMeOH + 0.1 M NaCl | −0.6 V vs. Ag/AgCl | AA2024 aluminum alloy | 1% γ-APS-doped and 2.5% BTSE-doped epoxy coating | Pt (25) | [129] |
| $O_2$ | 0.1 M KCl | −0.70 V vs. Ag/AgCl/KCl sat. | CS | Two-component polyurethane film | Pt (10) | [166] |
| $O_2$ | 0.9 mM FcMeOH + 5 wt.% NaCl | −0.7 V vs. Ag/AgCl/KCl sat. | Q235 MS | Enamel coating | – | [130] |
| $O_2$ | 0.5 mM FcMeOH + 3.5 wt.% NaCl | −0.7 V vs. Ag/AgCl/KCl sat. | High strength steel (SAPH440) | Enamel coating | – | [131] |
| $O_2$ | 5 mM FcMeOH + 0.05 M NaCl | −0.6 V vs. Ag/AgCl/KCl sat. | AA2024-T3 | Epoxy coating and epoxy coating containing silyl-ester doped capsules | Pt (5) | [132] |
| $O_2$ | 1 mM $K_4[Fe(CN)_6]$ + 0.1 M NaCl | −0.7 V vs. Ag/AgCl/KCl sat. | GS | Two-component epoxy primer containing zinc phosphate | Pt (10) | [152] |
| $O_2$ | 3 wt.% NaCl | −0.65 V vs. Ag/AgCl/KCl (3 M) | CS | Polypyrrole | Pt (25) | [201] |
| $O_2$ | 0.2 M $H_3BO_3$ + 0.05 M BBS + 3.5 wt.% NaCl | −0.642 V vs. SCE | 304 SS | CrN film | Pt (15) | [202] |
| $O_2$ | 0.05 M NaCl | −0.6 V vs. Ag/AgCl/KCl sat. | AA2024-T3 aluminum alloy | SMPU polymer containing 8% polyurethane | Pt (10) | [203] |
| $O_2$ | 10 mM NaCl | −0.65 V vs. Ag/AgCl/KCl (3 M) | Cu + Fe | BTAH | Pt (25) | [83] |
| $O_2$ | 1 mM FcMeOH + 0.2 M KCl | −0.7 V vs. Ag/AgCl/KCl sat. | Al flakes | PPy doped with either tungstate or vanadate | Pt (10) | [204] |
| $O_2$ | 1 mM FcMeOH + 0.2 M KCl | −0.7 V vs. Ag/AgCl | AA2024-T3 aluminum alloy | Epoxy resin + vanadate- and tungstate-doped PPy/Al flake composite pigments | Pt (10) | [139] |
| $O_2$ | 3.5 wt.% NaCl | −0.70 V vs. Ag/AgCl/KCl sat. | CS | Epoxy | Pt (10) | [186] |
| $O_2$ | 0.1 M NaCl | −0.70 V vs. Ag/AgCl/KCl sat. | High tensile strength (HT) steel | Epoxy | Pt (10) | [187] |
| $O_2$ | 3.5 wt.% NaCl | −0.75 V vs. SCE | Q235 MS | PANI + $TiO_2$ particles | Pt (10) | [205] |
| $O_2$ | 0.1 M NaCl | −0.70 V vs. Ag/AgCl | CS | Epoxy | Pt (10) | [206] |

| | | | | | | |
|---|---|---|---|---|---|---|
| O$_2$ | 1 M NaClO$_4$ + 1 mM HClO$_4$ | −0.7 V vs. Ag/AgCl/KCl (3 M) | Steel | Two-component epoxy | Pt (10) | [188] |
| O$_2$ | 3.5 wt.% NaCl | −0.75 V vs. SCE | Q235 CS | BTAH and a SMP | Pt (10) | [207] |
| O$_2$ | 3.5 wt.% NaCl | −0.7 V vs. Ag/AgCl/KCl (3 M) | Low-CS | NFC and MFC | Pt (10) | [208] |
| O$_2$ | 0.1 wt.% NaCl + DHS | −0.75 V vs. Ag/AgCl | AA 2024-T3 Aluminium alloy | Silane-modified multi-layer with Mg-rich pigment | Pt (10) | [209] |
| O$_2$ | DHS | −0.75 V vs. Ag/AgCl | AA 2024-T3 Aluminium alloy | Silane-modified multi-layer with Mg-rich pigment | Pt (10) | [210] |
| O$_2$ | 0.1 M KCl | −0.70 V vs. Ag/AgCl/KCl sat. | MS | Polyurethane | Pt (10) | [170] |
| O$_2$ | 0.1 M KCl + 0.1 M Na$_2$B$_4$O$_7$ | −0.6 V vs. Ag/AgCl/KCl sat. | CS | Polyurethane | Pt (10) | [199] |
| O$_2$ | 3.5 wt.% NaCl | −0.75 V vs. SCE | Q235 CS | Epoxy | Pt (25) | [211] |
| O$_2$ | Natural seawater | −0.70 V vs. Ag/AgCl | MS | DGEBA + CeO$_2$ nanoparticles | Pt (10) | [191] |
| O$_2$ | 0.1 M NaCl | −0.70 V vs. Ag/AgCl/KCl sat. | CS | Epoxy coatings containing magnesium nanoparticles | Pt (10) | [193] |
| O$_2$ | 5% NaCl | −0.70 V vs. Ag/AgCl/KCl sat. | CS | Zinc coating with SiO$_2$ nanoparticles | Pt (10) | [200] |
| O$_2$ | 0.5 mM FcMeOH + 5 wt.% NaCl | −0.70 V vs. Ag/AgCl/KCl sat. | CS | Silane-Zinc | Pt (10) | [212] |
| O$_2$ | 0.1 M NaCl | −0.70 V vs. Ag/AgCl/KCl sat. | Mn steel | Epoxy | Pt (10) | [194] |
| O$_2$ | 3.5 wt.% NaCl | −0.75 V vs. SCE | Q235 CS | Waterborne epoxy resin + ATP nanoparticles | − | [213] |
| O$_2$ | 0.1 M KCl, 0.1 M Na$_2$SO$_4$ and 0.1 M Na$_2$B$_4$O$_7$ | −0.60 V vs. Ag/AgCl/KCl sat. | CS | Two-component epoxy-polyamine film containing glass flake | Pt (10) | [214] |
| O$_2$ | 3.5 wt.% NaCl | −0.7 V vs. Ag/AgCl/KCl sat. | 7075 aluminum alloy | Berberine | Pt (10) | [172] |
| O$_2$ | 3.5 wt.% NaCl | −0.75 V vs. SCE | AA2024-T3 aluminum alloy | Shape memory epoxy polymers containing dual-function microspheres | Pt (−) | [215] |

## 3.4. Combined Operation Modes

The signal recorded by the sensing probes of SECM consists of a complex combination of spatially-resolved information originating from the distance between the

tip and the surface of the sample under study (i.e., sensitive to the morphology of the substrate) and of the actual chemical response due to the reactivity of the substrate, which in practice mainly limits its use to the characterization of flat surfaces and to the first stages of the formation of the surface film due to its progressive roughness with degradation. Although the morphological and chemical information may be ultimately convoluted in conventional SECM analysis, the first contribution can be considered constant in the case of a flat surface, so that changes in probe response can be attributed to the chemical reactivity of the studied system. Unfortunately, highly reactive systems occurring in light-weight alloy materials, which rapidly develop layers of oxide products and gas evolution under common operating conditions, and those associated with the self-healing mechanisms of smart coatings containing nanoreservoirs for functionalized operation, do not exhibit flat surfaces. This feature is not a real limitation for the investigation of thin films on metals because their width dimensions are often much smaller than the size of the scanning probe, but it can make it difficult to characterize larger surface defects such as crevices or heterogeneous regions extending over a large surface compared to the tip dimensions such as those formed in welds. Notwithstanding, efficient measurement strategies have been developed to construct SECM surface images by combining separate images of smaller regions [216].

To overcome this limitation imposed by the convolution of topographic information and chemical activity in the signal measured at the tip, a multi-scale electrochemical methodological procedure can be performed to deconvolute chemical information relevant to corrosion reactions and protection mechanisms in such complex systems can be performed [217]. Souto and coworkers developed a quite simple and systematic methodological procedure involving the combined use of various operation modes in amperometric SECM to study defects in organic coatings of width and depth dimensions greater than those of the tip [170], as illustrated in Figure 8.

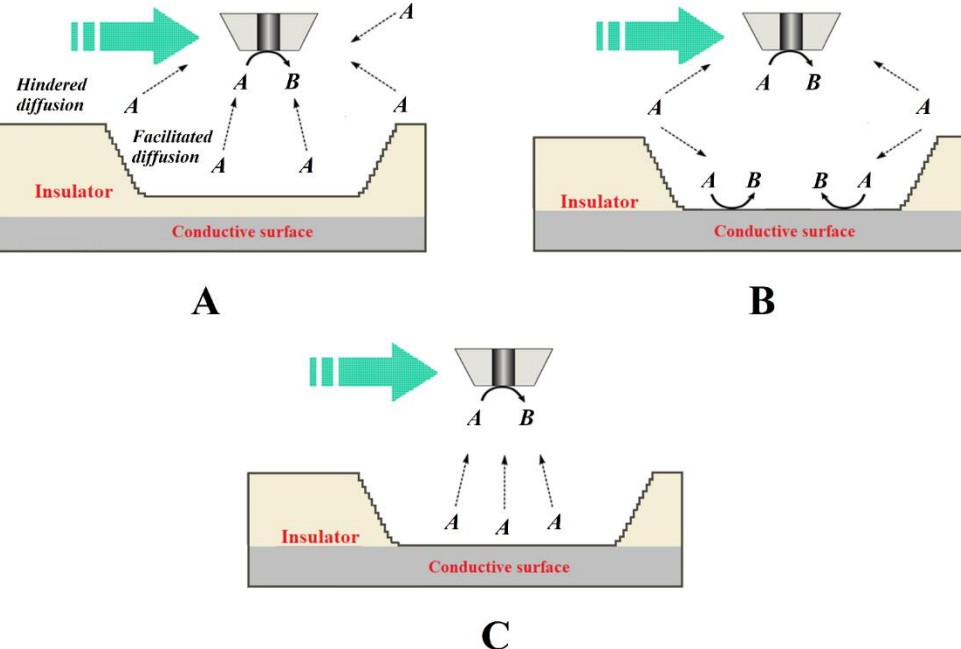

**Figure 8.** Diagrams of the processes that occur at the tip for the electroreduction of species *A* when the tip passes over a larger defect depending on whether it is an insulating or a conductive surface exposed to the aqueous medium. The green arrow indicates the scan direction. Depending on the source of species *A*, the following situations have been described: (**A**) species *A* is present in the aqueous medium and is not consumed within the defect, which behaves as an insulator; (**B**) species *A* is present in the aqueous medium and transforms at both the tip and the bottom of the defect; and (**C**) species *A* is generated at the bottom of the defect, although it was not originally present in the aqueous environment.

This procedure required the choice of detection modes more sensitive to either the topographical changes or the chemical activity by controlling the local chemistry of the system and the characteristics of the tip, followed by a subsequent stage of recording the combined signal of the complete corroding system. This methodology can be further extended by combining potentiometric modes with amperometric operation using multi-probe configurations, as recently demonstrated for the study of corrosion processes on cut edges of organically coated galvanized steel [42]. In this case, the combined amperometric/potentiometric SECM operation was performed by fabricating a multi-probe assembly using the same procedures previously developed for fabricating potentiometric probes with an internal reference electrode [40,120].

An alternate way to overcome the referred limitation is to associate SECM with other surface resolved techniques [218]. An option is the combination of SECM with AFM using cantilever probes modified for this purpose, which made it possible to simultaneously image the topography and the electrochemical activity in situ. In this way, the monitoring of nucleating corrosion pits in iron-based materials [219] and the dissolution-redeposition of metal ions in acidic environment [220] have been succesfully imaged in situ.

Besides the combination of SECM with topographically sensitive techniques, software routines can be designed so that the scan is actually performed at a constant tip-substrate distance (i.e., following the actual surface of the sample) instead of operating at a constant height, as is normally done in convential SECM operation [221]. Alternately, the measurement of shear forces between the tip and the surface can be used for constant distance operation in SECM instead of AFM [221]. The success of such an association was reported by Etienne et al. for monitoring the performance of self-healing coatings deposited on an aluminium alloy [222]. In their work, local features with a depth profile in excess of 50 μm were successfully resolved. In addition, local chemical analysis was simultaneously performed using in situ Raman spectroscopy.

Finally, corrosion reactions that progress far beyond their initial stages and eventually reach dimensions of a hundred micrometers or a few millimeters are already accessible using other experimental techniques and would not require the micrometric resolution of SECM.

## 4. Concluding Remarks

SECM is a powerful technique for the analysis and characterization of coatings of different nature applied over conductive and non-conductive substrates, as well as for self-healing coatings.

Three main operation modes are employed in amperometric SECM to this end, namely the feedback (both positive and negative feedback), generation/collection (G/C) mode (in either SG/TC or TG/SC configuration), and the redox competition modes. In some cases, various operation modes are used in the same experiment. Feedback mode is the most usual procedure, mainly due to the need of a controlled approach of the tip to the substrate.

The selection of the mediator depends on several factors, including the operation mode or the nature of the substrate and the solution employed. The most frequently employed mediators are ferrocene-methanol and hexacyanoferrate (III) in the feedback operation, iron in the G/C mode, and dissolved $O_2$ in redox competition mode. Pt disks of 10 and 25 μm of diameter are usually employed as tips. In minor proportion, Pt coated with parylene C, Pt/IrOx, Au and boron doped diamond tips are also used.

A rather recent addition to the characterization of surface layers and coatings applied on metals is the use of AC signals by using active MEs such as those employed in conventional amperometric operation of the SECM. The higher spatial resolution of the method, especially in the case of thin surface layers, makes these modes very attractive, in addition to the power of recording the images at different frequencies, thus allowing different processes to be distinguished in one measurement. Furthermore, by extracting the

electrochemical impedance from the localised potential and current signals, a frequency-resolved scanning electrochemical impedance microscopy (SEIM) technique is available.

Enhanced chemical selectivity is obtained by using micropotentiometric sensors in the SECM. Although their application was initially hindered by the significantly slower scanning rates that can be used due to the finite response times of ISMEs, which often compares poorly with the dynamics of the corrosion process under investigation, the development of new scanning and deconvolution methods are making this operation more attractive for the characterization of thin surface layers, as it already has with polymer coatings.

**Author Contributions:** Conceptualization, J.J.S.; investigation, J.J.S., J.I. and R.M.S.; data curation, J.J.S. and R.M.S.; methodology, J.I. and R.M.S.; writing—original draft preparation, J.J.S. and R.M.S.; writing—review and editing, J.J.S., J.I. and R.M.S.; visualization, J.J.S. and R.M.S.; funding acquisition, R.M.S. All authors have read and agreed to the published version of the manuscript.

**Funding:** The work was supported by the University of La Laguna and the Spanish Ministry of Science, Innovation and Universities (Madrid, Spain) under contract No. 2022/0000586.

**Institutional Review Board Statement:** Not applicable.

**Informed Consent Statement:** Not applicable.

**Data Availability Statement:** Not applicable.

**Acknowledgments:** J.J.S. acknowledges Géza Nagy for hosting his scientific mobility work to the University of Pécs (Hungary) from September to December, 2019, during which this work was designed and initiated.

**Conflicts of Interest:** The authors declare no conflict of interest.

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
