# Peer review of "Uses of Scanning Electrochemical Microscopy (SECM) for the Characterization with Spatial and Chemical Resolution of Thin Surface Layers and Coating Systems Applied on Metals: A Review"

_coatings, doi:10.3390/coatings12050637_

Round 1
Reviewer 1 Report
dear Author ,, it is good review in this field , it involved (( review about :the characterization with spatial and chemical resolution of thin surface layers and coating systems applied on metals))) , but it needs major corrections :
1- The introduction needs to clarification about corrosion on surface and wide explanation about this phenominon .
2- Also there is no any clarification about ({The reasons for not controlling the deep erosion that affects the surfaces in its depth? How can it be controlled chemically?
3- Clarify ((Figure 2. Typical cyclic voltammetry curve)). it is not clear .
4- I accept paper after major Corrections.
Author Response
1- The introduction needs to clarification about corrosion on surface and wide explanation about this phenominon .
Agreed. A new paragraph and one figure (new Figure 1) have been included in Section 1 (Introduction) to explain how corrosion processes are initiated on metals in aqueous environments, as well as to develop on the localized distributions of electrochemical microcells that develop in these cases.
2- Also there is no any clarification about ({The reasons for not controlling the deep erosion that affects the surfaces in its depth? How can it be controlled chemically?
A new paragraph has been included in Section 3 to describe the coupling of morphological and chemical resolved information in SECM and how to overcome the common limitation of imaging morphologically flat surfaces. The different scales of corrosion processes and examples of imaging with SECM corrosion processes that occur in depth have been included, providing 8 new references.
3- Clarify ((Figure 2. Typical cyclic voltammetry curve)). it is not clear .
The Reviewer has noted a misleading sentence in the mentioned figure caption. The rationale was to show the CV for ferrocene-methanol on Pt as this is redox mediator process that is most frequently used in SECM applications involving coatings and thin layers applied on metals. We have carefully modified both the legend and the manuscript text to avoid a misunderstanding.
Reviewer 2 Report
This review is an attempt to describe the uses of SECM for the characterization with spatial and chemical resolution of thin surface layers and coating systems applied on metals. The paper explains SECM ion general, and some references for surface coatings and thin surface layers on metals are included. The abstract mainly talks about UMEs and does not provide highlights of the review. Additionally, there is inadequate description of the surface coatings, reasons for the tips used or operation mode used, specific to coated metals. No discussion is provided for an altered/modified mechanism for SECM application on surface coatings on metals.
- The review explains the modes of operation of SECM in detail. But, the need for the review is not clearly understood. How is this review different that already available reviews on SECM? Line 108 says, “no attempt was made to distinguish between specific research interests like metallic corrosion or coatings, for example, whereas certain key experimental parameters of the measurements were missing such as the potential applied to the tip or the composition of the measuring solution.”. Although this is given to explain the “Novelty”, the review in itself does not discuss these features in detail. Further, no discussion is given on the SECM data from the literature obtained from different modes or different redox mediators.
- The review explains the modes of operation of SECM in detail, however fails to explain what happens with samples/thin surfaces with coatings or what different will appear between a coated and an uncoated sample. How do the modes of operation alter or are chosen when coated samples are used? How the species generated at the surface of coated sample at G/C mode differ from an uncoated sample?
- The title says, “Uses of scanning electrochemical microscopy (SECM) for the characterization with spatial and chemical resolution of thin surface layers and coating systems applied on metals”. However, the review just explains the modes of operation in an SECM with additional listing of redox mediators used for coated systems. Specific conditions used by the earlier literature to obtain spatial and chemical resolution have not been discussed. Moreover, a detailed section can be devoted to the tips used for coated samples. What modifications have been done in previous literature for the tips when applied to thin surface layers and coating systems applied on metals and why these modifications were necessary?
- In most operation modes, redox mediators are discussed. But, a reason explaining the need for a variable redox mediator is not explained. Line 433 explains the need for using O2 as a redox mediator. In a similar manner, literature discussion with the need to use/alter the redox mediator should be given.
- Literature is not discussed. Literature is cited but the research that has been done previously by other researchers, their thin surface layers/coated systems, and the mode operation, the tip used or the modifications to the tip used have not been discussed.
- Conclusions, Line 457: A rather recent addition to the characterization of surface layers and coatings applied on metals is the use of AC signals by using active UME such as those employed in conventional amperometric operation of the SECM. Literature discussion on what kinds of UMEs have been developed is required. Conclusions of the review do not go with the current review.
- The review is an attempt to describe SECM for thin surface layers and coatings on metals. Yet, no discussion on kind of coating is provided. Does the tip, parameter and mode of operation change if the metal coating is polymeric, organic, nanoparticle or nanocarbon based? How does the homogeneity or the thickness of the surface coating on the metal affect the SECM data and spatial resolution?
Formatting is required:
- Images quality in the entire review is poor. Even if adapted from literature or drawn by themselves, the quality of the images in the reviews should be improved.
- There are grammatical errors in the entire review. Please use proper English and avoid grammar mistakes.
- The structure loses connection. For example, from line 174 to 175 to 177. Line 175 breaks the connection and is out of place there. Several places in the review are like this.
- Line 290: “Although a detailed review on the use of potentiometric probes as SECM sensors can be found elsewhere [829] ...”. There is no reference numbered as 829. Please correct.
Restructuring is required:
- SECM instrumentation is not clearly described. Additionally, the section is not divided in a proper manner. As described, various SECM experimental procedures classified into amperometric and potentiometric operations. The section 2.0 is divided as Experimental design for SECM operation
2.1 SECM instrumentation
2.2 Tips used for amperometric operation
2.3 Redox mediators
2.4 Tips used for potentiometric operation
Are redox mediators not used for Potentiometric operation? Line 272 says, “In other cases, positioning is achieved with dual microelectrodes with the feedback mode by adding a redox mediator to the electrolyte”. Since, redox mediators are not specific to amperometric operation and can be used in both amperometric and potentiometric operation, they should be a separate category in section 2 after describing both amperometric and potentiometric operations.
- Section 2.3: This explains the importance of a redox mediator and how it is chosen. This is further illustrated with the help of examples in Table 2. The section clearly states, “The selection of a certain redox mediator for a given experiment is a critical issue for a successful experiment and depends on several factors, such as: • the nature of the sample studied; • the nature of the mediator. However, Table 2 does not include which thin surface or coating on what metal does the mediator refer to, alongwith the thickness of the metal coating.
- Coming to section 3, this again describes the operation modes in amperometric operation and the redox mediators used. Section and 2 and 3 both basically define the modes of operation, amperometric or potentiometric. These two sections should be combined and restructured in a readable, non-repeatable and clear format. Since several redox mediators are described here, they can be a separate section and can be explained and differentiated/tabulated on the basis of the operating modes used in a proper manner.
Author Response
This review is an attempt to describe the uses of SECM for the characterization with spatial and chemical resolution of thin surface layers and coating systems applied on metals. The paper explains SECM ion general, and some references for surface coatings and thin surface layers on metals are included. The abstract mainly talks about UMEs and does not provide highlights of the review. Additionally, there is inadequate description of the surface coatings, reasons for the tips used or operation mode used, specific to coated metals. No discussion is provided for an altered/modified mechanism for SECM application on surface coatings on metals.
1. The review explains the modes of operation of SECM in detail. But, the need for the review is not clearly understood. How is this review different that already available reviews on SECM? Line 108 says, “no attempt was made to distinguish between specific research interests like metallic corrosion or coatings, for example, whereas certain key experimental parameters of the measurements were missing such as the potential applied to the tip or the composition of the measuring solution.”. Although this is given to explain the “Novelty”, the review in itself does not discuss these features in detail. Further, no discussion is given on the SECM data from the literature obtained from different modes or different redox mediators.
The Reviewer is correct by pointing to a very unfortunate sentence in line 108 of the previous version of the manuscript, that has been modified as it follows:
“Unfortunately, no similar effort has been made in previous reviews on the application of SECM to the study of corrosion processes for covering key experimental parameters of the measurements for data such as the potential applied to the tip, the composition of the measuring solution, tip stability and dimensions, or the eventual effect of redox mediator conversion at the tip on the actual corrosion process under investigation.”
2. The review explains the modes of operation of SECM in detail, however fails to explain what happens with samples/thin surfaces with coatings or what different will appear between a coated and an uncoated sample. How do the modes of operation alter or are chosen when coated samples are used? How the species generated at the surface of coated sample at G/C mode differ from an uncoated sample?
These ideas have been expanded in the revised version of the manuscript.
3. The title says, “Uses of scanning electrochemical microscopy (SECM) for the characterization with spatial and chemical resolution of thin surface layers and coating systems applied on metals”. However, the review just explains the modes of operation in an SECM with additional listing of redox mediators used for coated systems. Specific conditions used by the earlier literature to obtain spatial and chemical resolution have not been discussed. Moreover, a detailed section can be devoted to the tips used for coated samples. What modifications have been done in previous literature for the tips when applied to thin surface layers and coating systems applied on metals and why these modifications were necessary?
Although not every citation has been discussed separately, the features requested by the Reviewer have been considered in their corresponding Sections, while describing their limitations and best practices as supported by the cited works that are considered to better describe their operation.
4. In most operation modes, redox mediators are discussed. But, a reason explaining the need for a variable redox mediator is not explained. Line 433 explains the need for using O2 as a redox mediator. In a similar manner, literature discussion with the need to use/alter the redox mediator should be given.
A new paragraph in Section 2.3 has been included regarding redox mediators. In addition, a new Figure 8 and Section 3.4, next to other smaller additions, have been included in this revision.
5. Literature is not discussed. Literature is cited but the research that has been done previously by other researchers, their thin surface layers/coated systems, and the mode operation, the tip used or the modifications to the tip used have not been discussed.
Although not every citation has been discussed separately, the features requested by the Reviewer have been considered in their corresponding Sections, while describing their limitations and best practices as supported by the cited works that are considered to better describe their operation.
6. Conclusions, Line 457: A rather recent addition to the characterization of surface layers and coatings applied on metals is the use of AC signals by using active UME such as those employed in conventional amperometric operation of the SECM. Literature discussion on what kinds of UMEs have been developed is required. Conclusions of the review do not go with the current review.
The issue is developed more extensively in the new Section 3.4. It must be noticed that the nature and geometry of the amperometric tip in a multi-barrel (i.e., combined) arrangement are precisely the same as in conventional single-tip applications.
7. The review is an attempt to describe SECM for thin surface layers and coatings on metals. Yet, no discussion on kind of coating is provided. Does the tip, parameter and mode of operation change if the metal coating is polymeric, organic, nanoparticle or nanocarbon based? How does the homogeneity or the thickness of the surface coating on the metal affect the SECM data and spatial resolution?
When differences are required, they have been described in the manuscript, whereas those operations that can be done with the same arrangement are described for all the systems covered in the review without mentioning specific cases.
Formatting is required:
1. Images quality in the entire review is poor. Even if adapted from literature or drawn by themselves, the quality of the images in the reviews should be improved.
All the figures have been edited.
2. There are grammatical errors in the entire review. Please use proper English and avoid grammar mistakes.
The text has been revised as indicated.
3. The structure loses connection. For example, from line 174 to 175 to 177. Line 175 breaks the connection and is out of place there. Several places in the review are like this.
Revised
4. Line 290: “Although a detailed review on the use of potentiometric probes as SECM sensors can be found elsewhere [829] ...”. There is no reference numbered as 829. Please correct.
Corrected. In the previous version of the manuscript it was meant to say [82].
Restructuring is required:
1. SECM instrumentation is not clearly described. Additionally, the section is not divided in a proper manner. As described, various SECM experimental procedures classified into amperometric and potentiometric operations. The section 2.0 is divided as Experimental design for SECM operation
2.1 SECM instrumentation
2.2 Tips used for amperometric operation
2.3 Redox mediators
2.4 Tips used for potentiometric operation
Done accordingly
2. Are redox mediators not used for Potentiometric operation? Line 272 says, “In other cases, positioning is achieved with dual microelectrodes with the feedback mode by adding a redox mediator to the electrolyte”. Since, redox mediators are not specific to amperometric operation and can be used in both amperometric and potentiometric operation, they should be a separate category in section 2 after describing both amperometric and potentiometric operations.
Redox mediators are not used in normal potentiometric operation. A different situation has arisen by the work of our group when developing multiprobe arrangements that accommodate amperometric and potentiometric tips in a single probe. In that situation, the operation of the amperometric tip in the arrangement is precisely the same as in the conventional single tip arrangement and it is not necessary to introduce a new subsection. Anyway, the text was clarified to explain this situation
3. Section 2.3: This explains the importance of a redox mediator and how it is chosen. This is further illustrated with the help of examples in Table 2. The section clearly states, “The selection of a certain redox mediator for a given experiment is a critical issue for a successful experiment and depends on several factors, such as: • the nature of the sample studied; • the nature of the mediator. However, Table 2 does not include which thin surface or coating on what metal does the mediator refer to, along with the thickness of the metal coating.
The different redox mediators are first described in Table 2, whereas the actual systems and applications are described in the Tables associated with Section 3. In the latter, details are also given regarding the nature and size of the tip, the composition of the test environments. In this way, the Reader can easily detect whether it corresponds to a coating layer or a thin surface film.
4. Coming to section 3, this again describes the operation modes in amperometric operation and the redox mediators used. Section and 2 and 3 both basically define the modes of operation, amperometric or potentiometric. These two sections should be combined and restructured in a readable, non-repeatable and clear format. Since several redox mediators are described here, they can be a separate section and can be explained and differentiated/tabulated on the basis of the operating modes used in a proper manner.
The proposal of the Reviewer was considered previously by the authors when organizing the manuscript among various organizations of the work, but we observed that the manuscript would focus too much on the conventional amperometric operation before allowing the reader to observe the variety of options available in addition to the conventional SECM usage.
Round 2
Reviewer 1 Report
yes now it is ok ,,, the author have done all corrections ,, now I accepted paper
Reviewer 2 Report
The pdf received by the reviewers have blurred images and Figures 5, 6 and 8 are out of the page. Please fix before final submission.
The revisions are fine and can be accepted in this form.